# When Hallucination Costs Millions: Benchmarking AI Agents in High-Stakes Adversarial Financial Markets

## Abstract

We present CAIA, a benchmark exposing **a critical blind spot in AI evaluation: the inability of state-of-the-art models to operate in adversarial, high-stakes environments where misinformation is weaponized and errors are irreversible**. While existing benchmarks measure task completion in controlled settings, real-world deployment demands resilience against active deception. Using cryptocurrency markets as a natural laboratory, where $30 billion was lost to exploits in 2024, we evaluate 17 leading models on 178 time-anchored tasks requiring agents to distinguish truth from manipulation, navigate fragmented information landscapes, and make irreversible financial decisions under adversarial pressure.

Our results reveal a fundamental capability gap: without tools, even frontier models achieve only 12-28% accuracy on tasks junior analysts routinely handle. Tool augmentation improves performance but plateaus at 67.4% (GPT-5) versus 80% human baseline, despite unlimited access to professional resources. Most critically, **we uncover a systematic tool selection catastrophe: models preferentially choose unreliable web search (55.5% of invocations) over authoritative blockchain data**, falling for SEO-optimized misinformation and social media manipulation. This behavior persists even when correct answers are directly accessible through specialized tools, suggesting foundational limitations rather than knowledge gaps.

The implications extend beyond cryptocurrency to **any domain where adversaries actively exploit AI weaknesses**, e.g. cybersecurity, content moderation, etc. Our finding that Pass@k metrics mask dangerous trial-and-error behavior challenges fundamental assumptions about autonomous deployment. We release CAIA with contamination controls and continuous updates, establishing adversarial robustness as a necessary condition for trustworthy AI autonomy. The benchmark reveals that current models, despite impressive reasoning scores, remain fundamentally unprepared for environments where intelligence must survive active opposition.

## 1 Introduction

**The Gap Between Benchmark Performance and Autonomous Agent Deployment.** Artificial intelligence benchmarks guide optimization, shape incentives, and define progress in modern AI Deng et al. (2009); Wang et al. (2019). Over the past year, foundation models have achieved remarkable milestones: OpenAI models won the International Collegiate Programming Contest Foundation (2025), and Gemini with DeepThink solved International Mathematical Olympiad problems at gold-medal level Luong & Lockhart (2025), surpassing most human experts. These achievements have fueled optimism about deploying autonomous AI agents with minimal human oversight. Yet this optimism rests on a dangerous assumption that high scores translate directly to real-world readiness.

Most benchmarks evaluate models in closed worlds where tools function as expected, information is trustworthy, and other agents cooperate Mialon et al. (2023); Zhou et al. (2023); Yao et al. (2025); Zheng et al. (2025). **They measure competence, not resilience.** Real-world autonomy requires surviving in open systems rife with uncertainty, misinformation, and adversarial incentives. Agents deployed in finance, governance, or infrastructure must distinguish truth from manipulation, avoid catastrophic failure, and act conservatively under uncertainty. Evaluation of autonomous AI agents, where trustworthy deployment is the top priority, should therefore critical capabilities explicitly.

This gap creates a perilous blind spot in measuring AI progress. An agent that excels on challenging reasoning benchmarks may still believe fabricated news, purchase compromised assets, or fall for phishing attacks, because nothing in its evaluation prepared it for deception. As AI agents increasingly interact with untrusted users, real money, and critical infrastructure, this vulnerability represents a safety concern hiding behind impressive scores Amodei et al. (2016); Hendrycks et al. (2021).

We argue for an opinion shift in agent evaluation: **Beyond measuring task completion on curated problems, evaluations should test robust survival in adversarial, high-stakes environments**. Rather than escalating difficulty alone, we should simulate hostile settings where others actively deceive, information is weaponized, and irreversible failures cause substantial loss. We introduce *CAIA*, the Crypto AI Agent Benchmark, which tests AI agent capabilities under these conditions.

**Crypto: A Natural Laboratory for Adversarial Robustness.** Cryptocurrency markets provide a unique environment for evaluating agent robustness under genuinely adversarial conditions. Despite controversy around speculation and fraud, these characteristics create ideal hostile testing conditions for AI agents. Crypto uniquely combines three properties essential for adversarial evaluation:

**1. Adversarial Environment with Sophisticated Deception.** The cryptocurrency ecosystem operates as a "dark forest" where misinformation is weaponized and adversaries actively hunt victims Robinson & Konstantopoulos (2020). Pseudonymous blockchains enable malicious actors to operate without reputation consequences. Potential profits motivate sophisticated attack strategies. Regulatory gaps permit deception tactics illegal in traditional markets. Daily occurrences include honeypot contracts designed to trap victims Hu et al. (2021), flash loan exploits manipulating prices within single transactions Cao et al. (2021), and coordinated social engineering campaigns Li et al. (2023). These real adversarial conditions require agents to genuinely distinguish truth from manipulation.

**2. High Stakes with Immediate Consequences.** Cryptocurrency markets lack traditional financial safeguards. Transactions are irreversible, smart contract executions are final, and no central authority can reverse fraudulent transfers. In 2024 alone, over $30 billion was lost to exploits and scams Chainalysis Team (2025). When an AI agent makes a tiny mistake, losses cannot be recovered. This creates genuine high-stakes conditions where errors have immediate, permanent monetary consequences and malicious actors are economically incentivized to exploit weaknesses.

**3. Transparent and Verifiable Ground Truth.** Despite adversarial chaos, cryptocurrency offers complete transparency and immutability. Every transaction, smart contract interaction, and token transfer is permanently recorded on public blockchains Zhang et al. (2021). This enables unique evaluation conditions where: (1) agent decisions can be verified against immutable on-chain records; (2) financial losses trace to specific transactions with cryptographic proof; (3) attack patterns can be analyzed retroactively with perfect information. This transparency in an adversarial environment enables reproducible evaluation with real-world relevance, addressing fundamental limitations of traditional financial benchmarks that must choose between proprietary data or synthetic simulations.

Current AI systems enter this domain fundamentally unprepared. Trained predominantly on centralized, indexed, trustworthy "Web2" data Common Crawl Foundation (2024); Dodge et al. (2021), they lack exposure to crypto's fragmented, rapidly-evolving "Web3" information landscape. Blockchain data spans thousands of nodes without central access points; DeFi protocols update daily without documentation; critical information exists in ephemeral social channels that evade crawlers Zhang et al. (2021). Even accessible content is often adversarial, consisting of deliberately misleading information, scams, and market manipulation. This combination makes crypto particularly challenging for AI trained on traditional web data, and hence an ideal testbed for AI agents' adversarial robustness.

**CAIA: Benchmarking Intelligence Under Fire.** We present CAIA (Crypto AI Agent Benchmark), the first benchmark explicitly designed to evaluate AI agents in an actively hostile, high-stakes environment. Unlike existing benchmarks measuring task completion in controlled settings, all tasks in CAIA are grounded in crypto, which measure survival and truth-seeking under adversarial pressure.

Our evaluation reveals significant gaps between state-of-the-art large language models and junior human analysts. Models achieve only 12-28% accuracy without tools. Even when equipped with tools providing correct answers, the accuracy at best is 67.4% (`GPT-5`), while entry-level human analyst baselines reach 80%. Models consistently rely on unreliable web search over domain-specific tools that directly link to the source of truth, suggesting fundamental limitations in tool selection

and adversarial reasoning. These patterns reveal that, when users entrust capital to autonomous agents expecting intelligent fund management, agents may effectively be guessing and attempting "trial-and-error", which is extremely dangerous in high-stakes adversarial scenarios.

**Our Contributions.** Our work advances agent evaluation through four primary contributions:

**Adversarial-First Evaluation:** While existing benchmarks assume cooperative environments, CAIA introduces active deception, source validation, and adversarial robustness as core capabilities, reflecting deployment reality where agents face hostile actors, not just noisy data.

**Financial Reality Grounding:** Using real market tasks where mistakes have monetary consequences creates accountability and real-world transferability absent from synthetic benchmarks.

**Temporal Precision Testing:** Time-anchored tasks evaluate multi-timescale reasoning and data obsolescence handling required in volatile markets, beyond static benchmark capabilities.

**Diagnostic Failure Analysis:** Fine-grained diagnostics of evaluation results provide actionable insights about specific failure modes, critical for both model development and deployment decisions.

The implications extend beyond crypto. As AI agents enter other adversarial domains, e.g. cybersecurity, content moderation, medical diagnosis, CAIA's measured capabilities become universally critical. Crypto represents an extreme adversarial environment characterized by pervasive misinformation, sophisticated scams, and active financial exploits. Success on CAIA therefore provides high confidence for autonomous deployment in any domain where adversaries actively exploit weaknesses, and establishes a strong foundation for routine deployment in less hostile environments.

**Paper Organization.** In the following, Section 2 presents CAIA's design philosophy and task curation methodology. Section 3 details our experimental framework and quantitative evaluation results across 17 state-of-the-art models. Section 4 analyzes failure modes and derives insights for improving and deploying AI agents. Section 5 discusses future directions and concludes the paper.

## 2 BENCHMARK CURATION

### 2.1 DESIGN PRINCIPLES

CAIA addresses a critical gap in agent evaluation: the absence of benchmarks that capture the worst-case performance under adversarial, high-stakes scenarios, which is exactly the nature of real-world crypto analysis. We identify three core challenges that define this domain:

- irreversible financial consequences where incorrect decisions lead to permanent capital loss (e.g., MEV and execution risks) Daian et al. (2019); Qin et al. (2021);

- an adversarial information landscape, including coordinated manipulation and pump-and-dump campaigns Xu et al. (2019); Ardia & Bluteau (2023);

- high-density, multi-source data that mixes on-chain traces, social signals, and protocol documentation Mialon et al. (2023); Zhou et al. (2023).

Our community-driven curation process, involving over 3,000 contributors including protocol developers, quantitative researchers, and venture capital investors, ensures ecological validity.

To mitigate training-data contamination, a persistent threat to static benchmarks Deng et al. (2024); Carlini et al. (2021), CAIA anchors tasks to recent market events with explicit temporal constraints (block heights, timestamps), following best practices from time-sensitive evaluation Chen et al. (2021); Kasai et al. (2023), and creating an evaluation framework resistant to memorization-based solutions. We will also actively retire out-dated tasks and add new tasks to ensure liveness.

### 2.2 QUALITY ASSURANCE

For each task in CAIA, quality is guaranteed through three foundational pillars that mirror expert analytical workflows. This approach moves beyond isolated capability testing to evaluate complete reasoning and acting chains Yao et al. (2022), ensuring that successful task completion requires:

**Knowledge:** Evaluates foundational understanding of crypto-native concepts, from AMM mechanics to governance structures, testing conceptual grasp rather than definitional recall.

**Planning:** Assesses strategic decomposition of complex questions into executable analytical workflows, requiring agents to specify tool selection and sequencing before execution Wei et al. (2022).

**Action:** Tests real-world execution using production APIs (Etherscan, CoinGecko, DefiLlama) Team (2025c;a;b) dedicated for on-chain detection, evaluating both technical competence and judgment under realistic constraints like rate limiting and data inconsistency.

This progression from understanding through planning to execution reflects established cognitive architectures in complex problem-solving Newell (1990) specifically adapted for the crypto domain's unique requirements. Tasks that satisfy these conditions are naturally suited for testing AI agents' capabilities, as they precisely mirror the desired approach to tackling complex problems.

## 2.3 CURATION PIPELINE

Our dataset originates from more than 10,000 authentic queries we collect from over 3,000 active users spanning different roles, representing the largest and most comprehensive collection of real-world crypto analytical needs to date. Through a rigorous five-stage curation pipeline that operationalizes our design principles and quality metrics (Figure 1), we distill the candidate pool into 178 high-quality CAIA benchmark tasks, as detailed below:

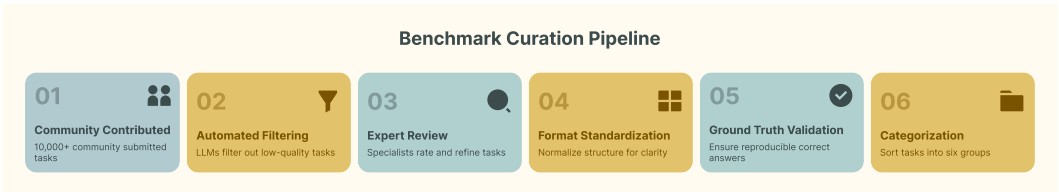

Figure 1: Data Curation Pipeline.

**Stage 1: Automated Filtering:** We apply the standard "LLM-as-a-judge" technique, using LLMs to filter out off-topic, ambiguous, non-answerable, or trivial queries while enforcing temporal grounding. After filtration, we ask LLMs to rate each task based on our quality assurance criteria in 2.2, retaining only the top 15%. This reduces the corpus size to approximately 1,000 tasks.

**Stage 2: Expert Review:** This stage mirrors the traditional paper reviewing process. Our expert team comprises 92 domain specialists, with each assigned 50 tasks to review and grade based on the quality assurance criteria in 2.2. Each task surviving Stage 1 receives at least 4 reviews, and we calculate the final score by averaging all reviews after removing the highest and lowest scores. The top 200 tasks advance to the final pool. After deduplicating similar tasks (i.e., handpicking tasks requiring similar execution logic), we obtain a prototype candidate set of 186 tasks.

**Stage 3: Format Standardization:** To address inconsistencies arising from different tones and writing styles, we unify the format of each task. This requires explicit anchoring to block numbers or timestamps, enabling objective evaluation and straightforward verification of ground truth answers.

**Stage 4: Ground Truth Validation:** For each task and its corresponding answer, we verify that a reproducible ground truth toolchain calling scheme exists and associate it with the task. We omit tasks that cannot be reproduced from the benchmark to ensure objectivity. This process provides much more than a single correct answer - It demonstrates the precise methodology that agents should follow to reach the correct solution, ensuring the accuracy, objectivity, and reproducibility of the desired answer. After this validation, we arrive at our final CAIA benchmark of 178 tasks.

**Stage 5: Categorization:** For diagnostics of model capabilities, we identify 6 fine-grained categories encompassing all tasks and carefully categorize each task accordingly, as shown in Table 1. This step enables detailed assessment beyond aggregate metrics, supporting our analysis in Section 4.

By design, CAIA addresses weaknesses noted in prior evaluations: contamination in static datasets Deng et al. (2024), lack of ecological validity in synthetic tasks Zhou et al. (2023); Mialon et al. (2023), and single-metric reporting that masks capability gaps Liang et al. (2022). Grounding

| Category | N | % | Focus | Validation Method |
|---|---|---|---|---|
| On-Chain Analysis | 77 | 43.3 | Transaction patterns, MEV, fund flows | Transaction hash verification |
| Project Discovery | 49 | 27.5 | Protocol evaluation, security analysis | Documentation cross-reference |
| Tokenomics | 23 | 12.9 | Incentive design, value accrual | Mathematical proof |
| Overlap | 14 | 7.9 | Multi-domain synthesis | Composite verification |
| Trend Analysis | 8 | 4.5 | Temporal patterns, adoption metrics | Statistical validation |
| General Knowledge | 7 | 3.9 | Foundational concepts | Canonical reference |

Table 1: Distribution of 178 benchmark tasks across 6 analytical categories.

in real-world high-stakes needs, operating under an adversarial environment with false information, verifiable with objective and immutable answers, CAIA provides a durable foundation for measuring autonomous agentic intelligence in adversarial financial markets.

## 3 EVALUATION RESULTS

### 3.1 EXPERIMENTAL SETUP

We conduct a comprehensive evaluation of 17 state-of-the-art large language models on the CAIA benchmark, encompassing leading proprietary models (`GPT-4.1`, `GPT-5`, `Claude`, `Gemini`, `Grok`, `Kimi`) and prominent open-source flagships (`Llama`, `Qwen`, `DeepSeek`, `GPT-OSS`).

**Tool Augmentation.** We evaluate each model under two distinct conditions that mirror complementary aspects of real-world deployment. The **without-tools** condition functions as a closed-book examination, testing models' internalized knowledge and reasoning capabilities when forced to rely solely on parametric memory. This reveals their fundamental understanding of concepts, market dynamics, and analytical reasoning without external assistance. Conversely, the **with-tools** condition resembles an open-book examination and tests agentic abilities where models gain access to 23 specialized tools spanning web search APIs, blockchain analytics platforms, market data feeds, and computational interpreters. Crucially, our data curation process in 2 ensures that correct answers are always accessible through appropriate tool use, and thus the challenge lies not in information availability but in tool selection and synthesis. This design choice deliberately isolates the agent's tool orchestration capabilities from knowledge limitations, providing a pure test of whether agents can identify and invoke the right resources when given unlimited access to professional instruments.

**Agentic Framework.** When equipped with tools, each model operates within a standardized ReAct-style Yao et al. (2022) agentic framework that handles tool dispatch, result parsing, and iterative reasoning. This ensures that our evaluation result is not affected by implementation variations.

**Human Baseline.** To establish human performance benchmarks, we recruited 16 participants from university blockchain clubs and early-stage blockchain companies, representing entry-level analyst expertise. These participants completed a stratified 10% sample of our benchmark, carefully balanced across all six analytical domains. Their averaged performance of 80% accuracy provides a critical baseline—notably, these junior analysts achieved this score in the open-book equivalent condition with full tool access, establishing the minimum bar for professional competence in the domain.

### 3.2 QUANTITATIVE PERFORMANCE ANALYSIS

To ensure robust evaluation, we employ multiple complementary metrics. Our primary measure is average accuracy via majority voting across five independent runs, which mitigates the substantial variance inherent in single-run evaluations of stochastic language models. We additionally report standard Pass@1 and Pass@5 metrics to capture both first-attempt performance and eventual success rates through exploration. However, as we will discuss in Section 4, the traditional Pass@k is misleading in high-stakes adversarial contexts where trial-and-error carries unacceptable risks.

Beyond performance metrics, we track computational costs by logging token consumption for each query and computing the associated monetary expense. This enables us to derive cost efficiency (cost-per-accuracy-point), revealing critical trade-offs between model capability and economic viability.

This analysis proves particularly illuminating when comparing proprietary APIs against open-source alternatives, where we observe up to 100-fold differences in cost for comparable performance.

Our results reveal a stark performance landscape that challenges fundamental assumptions about tool-augmented language models. As illustrated in Figure 2 and detailed in Tables 2 and 3, model performance exhibits a bimodal distribution heavily dependent on tool availability.

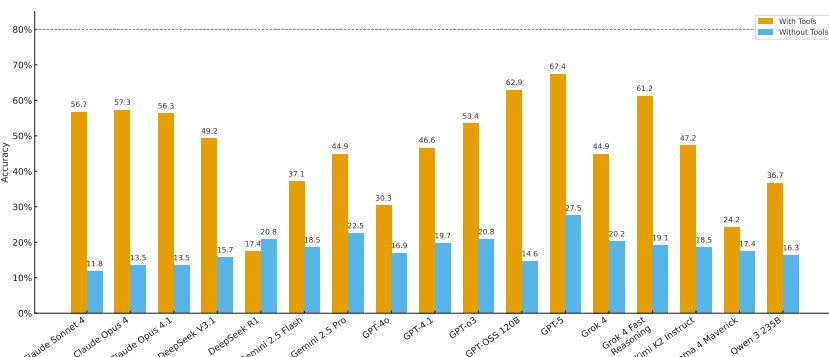

Figure 2: Average accuracy across five evaluation runs using majority voting. The dashed line indicates 80% human baseline performance. Without tools, all models perform near random (12–28%); with tools, performance improves but plateaus below human capability.

In the absence of external tools, every evaluated model, including frontier systems, demonstrates catastrophic failure, achieving merely 12–28% accuracy. This represents performance scarcely above random guessing for many tasks. Even `GPT-5`, the strongest model, has only 27.5% accuracy, indicating how poorly current parametric knowledge transfers to specialized adversarial domains.

On the other hand, tool augmentation yields substantial improvements, yet even our best-performing model `GPT-5` achieves only 67.4% accuracy, falling significantly short of the 80% human baseline established by junior analysts. This performance ceiling persists despite unlimited access to professional-grade tools and comprehensive documentation, suggesting fundamental architectural limitations of the state-of-the-art LLMs today, rather than simple knowledge gaps.

The cost-efficiency analysis reveals a striking economic disparity across model families. While proprietary systems like `Claude Opus 4` incur costs exceeding $1 per problem, open-source alternatives such as `GPT-OSS 120B` achieve competitive accuracy at under $0.01 per query, which is a remarkable 100-fold improvement in cost efficiency. Even more compelling, `GPT-OSS 120B` actually *outperforms* several proprietary models while maintaining this dramatic cost advantage. This economic reality has profound implications for deployment at scale: organizations processing thousands of queries daily could achieve near-frontier performance at a fraction of the cost, fundamentally challenging the assumed superiority of commercial APIs in specialized domains, as illustrated in 3.

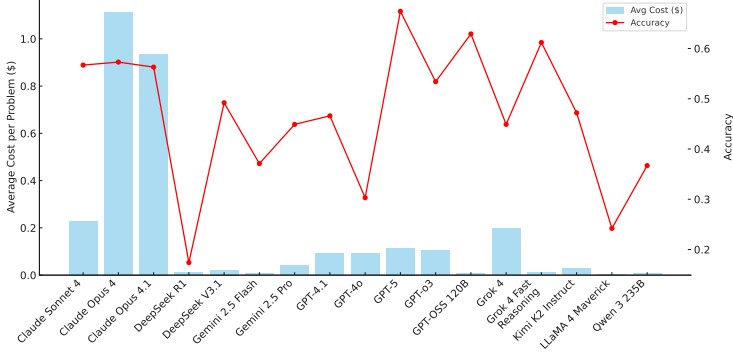

Figure 3: Cost-accuracy tradeoff reveals GPT-OSS 120B and Grok 4 Fast as Pareto-optimal choices, achieving near-frontier performance at minimal cost.

| Model | Majority Vote | Pass@1 (%) | Pass@5 (%) | Avg Cost ($) | Cost/Score |
|---|---|---|---|---|---|
| Claude Sonnet 4 | 0.118 | 12.9 | 18.0 | 0.0070 | 0.0593 |
| Claude Opus 4 | 0.135 | 13.5 | 16.9 | 0.0334 | 0.2481 |
| Claude Opus 4.1 | 0.135 | 15.2 | 17.4 | 0.0356 | 0.2642 |
| DeepSeek R1 | 0.208 | 21.9 | 35.4 | 0.0038 | 0.0184 |
| DeepSeek V3.1 | 0.157 | 15.7 | 29.2 | 0.0005 | 0.0030 |
| Gemini 2.5 Flash | 0.185 | 20.2 | 21.9 | 0.0012 | 0.0062 |
| Gemini 2.5 Pro | 0.225 | 20.2 | 29.8 | 0.0051 | 0.0226 |
| GPT-4.1 | 0.197 | 20.8 | 24.2 | 0.0025 | 0.0126 |
| GPT-4o | 0.169 | 19.1 | 20.8 | 0.0016 | 0.0098 |
| GPT-5 | 0.275 | 28.1 | 42.7 | 0.0207 | 0.0753 |
| GPT-o3 | 0.208 | 22.5 | 29.2 | 0.0085 | 0.0407 |
| GPT-OSS 120B | 0.146 | 18.5 | 21.3 | 0.0003 | 0.0022 |
| Grok 4 | 0.202 | 20.2 | 24.2 | 0.0345 | 0.1705 |
| Grok 4 Fast Reasoning | 0.191 | 21.3 | 23.6 | 0.0006 | 0.0029 |
| Kimi K2 Instruct | 0.185 | 17.4 | 25.3 | 0.0006 | 0.0033 |
| Llama 4 Maverick | 0.174 | 16.9 | 24.7 | 0.0003 | 0.0015 |
| Qwen 3 235B | 0.163 | 14.6 | 18.0 | 0.0010 | 0.0061 |

Table 2: Performance *without tools*: accuracy, Pass@k, cost, and cost efficiency across all models.

| Model | Majority Vote | Pass@1 (%) | Pass@5 (%) | Avg Cost ($) | Cost/Score |
|---|---|---|---|---|---|
| Claude Sonnet 4 | 0.567 | 57.9 | 66.9 | 0.2291 | 0.4037 |
| Claude Opus 4 | 0.573 | 59.6 | 71.9 | 1.1139 | 1.9439 |
| Claude Opus 4.1 | 0.563 | 56.3 | 69.0 | 0.9357 | 1.6614 |
| DeepSeek R1 | 0.174 | 26.4 | 54.5 | 0.0121 | 0.0695 |
| DeepSeek V3.1 | 0.492 | 55.9 | 71.2 | 0.0216 | 0.0438 |
| Gemini 2.5 Flash | 0.371 | 39.3 | 62.4 | 0.0070 | 0.0190 |
| Gemini 2.5 Pro | 0.449 | 49.4 | 61.2 | 0.0407 | 0.0906 |
| GPT-4.1 | 0.466 | 51.7 | 60.7 | 0.0913 | 0.1958 |
| GPT-4o | 0.303 | 50.0 | 55.6 | 0.0909 | 0.2997 |
| GPT-5 | 0.674 | 70.2 | 77.0 | 0.1154 | 0.1712 |
| GPT-o3 | 0.534 | 59.6 | 73.6 | 0.1047 | 0.1962 |
| GPT-OSS 120B | 0.629 | 56.2 | 72.5 | 0.0066 | 0.0104 |
| Grok 4 | 0.449 | 52.2 | 66.9 | 0.1980 | 0.4405 |
| Grok 4 Fast Reasoning | 0.612 | 57.9 | 71.9 | 0.0098 | 0.0160 |
| Kimi K2 Instruct | 0.472 | 46.6 | 64.6 | 0.0273 | 0.0579 |
| Llama 4 Maverick | 0.242 | 30.3 | 64.6 | 0.0031 | 0.0129 |
| Qwen 3 235B | 0.367 | 38.4 | 61.0 | 0.0062 | 0.0170 |

Table 3: Performance *with tools*: accuracy, Pass@k, cost, and cost efficiency across all models.

# 4 ANALYSIS AND DISCUSSION

## 4.1 THE ILLUSION OF COMPETENCE: WHY PASS@K METRICS MISLEAD

A critical finding emerges from the stark divergence between Pass@1 and Pass@5 metrics shown in Table 3. While Pass@1 accuracy hovers around 50–60% for top models except for 70.2% of `GPT-5`, Pass@5 consistently exceeds 60%, with `GPT-5` reaching 77%. This improvement might initially suggest robust problem-solving capabilities. However, this interpretation fundamentally misunderstands the nature of real-world deployment, particularly in high-stakes financial contexts.

Consider the implications: `Gemini 2.5 Flash`'s jump from 39.3% (Pass@1) to 62.4% (Pass@5) indicates that the model essentially *guesses* correctly through repeated trials rather than reasoning strategically on first attempt. In cryptocurrency markets, where a single incorrect transaction can result in irreversible financial loss, this trial-and-error approach represents an unacceptable risk profile. **Real-world financial decisions do not offer multiple attempts.** When users entrust capital to autonomous agents, expecting them to manage funds "cleverly" through clear reasoning, it is unacceptable and extremely dangerous if the agent is effectively guessing its next action.

The minimal improvement (and sometimes decrease) from Pass@1 to majority voting further underscores this concern. If models were exhibiting genuine understanding with occasional errors, we would expect majority voting to substantially improve accuracy. Instead, the modest gains suggest that errors stem from fundamental reasoning failures rather than stochastic variations. This pattern is particularly alarming given that automated agents are increasingly deployed in financial contexts where users may place unwarranted trust in their recommendations without human oversight.

## 4.2 DOMAIN-SPECIFIC TOOL USAGE AND PERFORMANCE PATTERNS

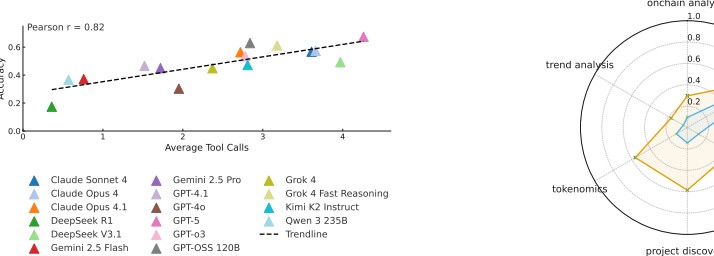
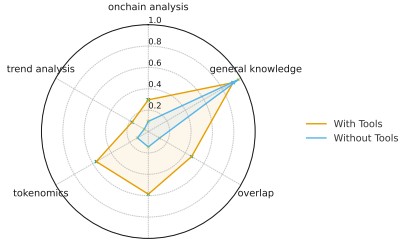

Figure 4: Tool usage frequency vs. accuracy.    Figure 5: Performance on different categories.

Figure 4 reveals a positive correlation between tool usage and accuracy with a Pearson coefficient of 0.82, showing that statistically more tool calls can help iterate and refine the response. On the other hand, the improvement by more tool calls does not persist at higher volumes (See Appendix A.2.) . Aggregated across all models, mean accuracy begins to decline once total tool calls exceed four, which suggests that effective tool use depends on strategic selection rather than quantity. Models making numerous unfocused tool calls may perform worse than those making fewer, well-targeted queries to appropriate tools.

Figure 5 shows that tool effectiveness varies substantially across CAIA's six categories. In **general knowledge**, tools offer little benefit because this information is already well represented in pre-training corpora. In contrast, **on-chain analysis** and **trend analysis** remain the lowest-performing categories despite modest gains from tool access, indicating that their difficulty stems from dynamic, data-dependent reasoning rather than missing information.

The intermediate categories—**tokenomics**, **project discovery**, and **overlap**—show the strongest tool-driven improvements, largely because they rely on information underrepresented in pre-training data, making targeted retrieval particularly valuable.

As we show next, however, these improvements are driven mainly by generic web search rather than specialized tools, reflecting that these domains face fewer incentives for targeted manipulation.

## 4.3 THE TOOL SELECTION CATASTROPHE

From our evaluation, we identify the systematic failure of models to select appropriate tools, even when optimal choices are unambiguous. Table 4 reveals that models default to generic web search for 55.5% of all tool invocations (combining Google and Twitter searches), despite having access to specialized blockchain analytics tools that provide authoritative data and direct answer.

| Tool Category | Invocations | Percentage |
|---|---|---|
| Google search | 11,626 | 49.6% |
| Specialized blockchain tools | 8,351 | 35.6% |
| URL fetching | 1,743 | 7.4% |
| Twitter search | 1,388 | 5.9% |
| Code execution | 355 | 1.5% |
| **Total** | 23,463 | 100.0% |

Table 4: Tool usage distribution reveals heavy reliance on generic search over domain-specific tools.

This behavior pattern is not merely suboptimal: it is dangerous. In an adversarial and manipulated environment, web search returns manipulated social media posts, coordinated shilling campaigns, and deliberately false information. Meanwhile, blockchain data provides immutable, verifiable ground truth. Yet models consistently choose the unreliable source over the authoritative one.

Our analysis reveals that certain tools require orchestration to be effective. Twitter search accuracy plummets from 40.7% when used in combination to 6.6% when used alone, indicating that social sentiment tools need market context to provide value. Conversely, direct blockchain queries (e.g., ERC-20 token info) maintain high accuracy in isolation. Models fail to recognize these compositional requirements, treating all tools as functionally equivalent, and problematically have a preference towards generic search tools, which may deliver second-hand manipulated information, over domain-specific tools that directly provide the source of truth for each query.

**A Case Study: When Simple Tasks Become Impossible.** Task 49 in CAIA epitomizes the depth of model failure in tool selection. The task requires retrieving monthly token launch counts from Pump.fun. The data is readily available through a single blockchain analytics API call, and the ground truth solution is trivial: DEFILLAMA_PUMP_STATS(MONTH="2025-01", METRIC="LAUNCHES")

Yet across all 17 evaluated models, **not a single one succeeded**. Instead, we observed a consistent pattern of cascading failure where models fall for misinformation:

1. Initial web searches return SEO-optimized but outdated blog posts
2. Refined searches for specific months yield social media speculation rather than data
3. Desperation leads to Twitter searches, surfacing coordinated misinformation
4. Models synthesize incorrect answers from these unreliable sources

The models never attempt to use DeFiLlama, Dune Analytics, or any blockchain-specific tool, despite these being explicitly documented and available. This represents not just a failed execution but a fundamental inability to recognize when specialized tools are necessary and identify source of truth.

## 5 CONCLUSION

We introduce CAIA, the first benchmark evaluating AI agents in high-stakes, adversarial environments. Our evaluation of 17 state-of-the-art models reveals critical gaps: leading models achieve only 67.4% accuracy with tools versus 80% human baseline, consistently preferring unreliable web search over specialized blockchain tools. The key obstacle is not tool access but fundamental lack of skeptical reasoning. Agents are easily misled by manipulation and confidently hallucinate critical data.

These vulnerabilities extend beyond crypto to any adversarial domain where misinformation is weaponized. Current models remain dangerously unreliable when stakes are high and adversaries are present. For trustworthy autonomy, future work should prioritize adversarial robustness over task completion metrics, and CAIA provides a vital testbed for building truly reliable autonomous agents.

## USE OF LARGE LANGUAGE MODELS (LLMS)

We used ChatGPT 5 and Claude Opus 4.1 exclusively for language polishing.

No model outputs were accepted without author verification.

## ETHICS STATEMENT

We affirm adherence to the ICLR Code of Ethics. Our benchmark curates tasks from public, verifiable sources in cryptocurrency markets, where ground truth can be checked against immutable on-chain records and production analytics APIs; no personal or proprietary user data are included in the released tasks. Human subjects research was limited to a small baseline study with entry-level analysts; we report only aggregate performance and do not release any individual-level data. We release CAIA openly with contamination controls and continuous updates, and commit to responsible handling of potentially sensitive insights.

## REPRODUCIBILITY STATEMENT

We take reproducibility seriously and provide all components necessary to independently re-create our results. The anonymized dataset is open to download and use at `https://huggingface.co/datasets/anonymous38527/anonymous_dataset` and will be continuously updated with new time-anchored tasks, and the anonymized repository with the artifacts for evaluation is open-sourced at `https://anonymous.4open.science/r/ICLR-1EA3`.

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

APPENDIX

# A ADDITIONAL EXPERIMENTS

## A.1 ERROR BAR ANALYSIS

To assess statistical uncertainty in model performance, we compute 95% bootstrap confidence intervals for every model–mode pair in CAIA. For each setting, we sample with replacement from the 178 benchmark items to generate 1 000 bootstrap replicates and estimate the distribution of mean accuracy.

**Tool-enabled models.**   High-performing tool modes, such as gpt_5 (0.674; 95% CI [0.601, 0.736]) and gpt_oss_120b (0.629; CI [0.562, 0.702]), form a clearly separated upper band. Their intervals sit 30-40 percentage points above their non-tool counterparts, indicating a large and statistically robust dependence on tool access.

**Non-tool models.**   The strongest non-tool modes, including gpt_5 (0.275; CI [0.213, 0.343]) and gpt_o3 (0.208; CI [0.152, 0.264]), cluster near the lower range of the benchmark. Their confidence intervals remain well below 0.5, reinforcing that CAIA tasks cannot be reliably solved without external tools.

**Mid-tier tool models.**   Models such as gemini_2.5_pro, gpt_4.1, grok_4, and kimi_k2 fall within an overlapping accuracy band of roughly 0.37–0.53. Their intervals substantially overlap, implying that fine-grained ordering within this tier is statistically inconclusive.

**Lower-tier tool models.**   Although models like llama_4 (0.242 mean accuracy) represent the lower end of tool-enabled performance, their intervals still exceed those of many non-tool modes. This further highlights the structural performance gap between tool and non-tool configurations.

**Summary.**   Overall, the bootstrap analysis confirms two key patterns: (1) tool use is the dominant driver of performance on CAIA, and (2) many model families with similar means have overlapping confidence intervals, underscoring the importance of uncertainty-aware comparisons rather than raw accuracy alone.

## A.2 TOOL CALL VOLUME AND ACCURACY ANALYSIS

We analyze how the number of external tool calls relates to task performance in CAIA. Although CAIA imposes no limit on tool usage, tool-enabled models naturally converge to a small number of calls per question. The results below summarize patterns observed across 15 130 tool-enabled items.

### A.2.1 RELATIONSHIP BETWEEN TOOL COUNT AND ACCURACY

Table 5 reports accuracy as a function of the number of distinct tools invoked per question. Single-tool usage yields the highest accuracy (mean 0.671), consistent with the fact that many retrieval-oriented tasks require only one correctly chosen tool. Accuracy decreases when models require multiple tools, reflecting the increased difficulty of such questions.

Table 5: Accuracy as a function of the number of distinct tools used per question.

| Tools Used per Question | Mean Accuracy | Samples |
|---|---|---|
| 0 | 0.192 | 3 545 |
| 1 | 0.671 | 3 575 |
| $\geq 2$ | 0.499 | 8 010 |

Table 6 extends this analysis to total call count (including repeated calls to the same tool). Accuracy peaks when models issue two to four calls and declines when five or more calls are used, suggesting diminishing returns and increasing uncertainty-driven exploration rather than productive reasoning.

Table 6: Accuracy as a function of total external tool calls.

| Total Tool Calls | Mean Accuracy | Samples |
|---|---|---|
| $\leq 1$ | 0.269 | 7 120 |
| 2 | 0.583 | 1 935 |
| 3–4 | 0.610 | 2 415 |
| $\geq 5$ | 0.382 | 3 660 |

Across all models, the correlation between final accuracy and either tool count or total call volume is weak (Spearman $\rho \approx 0.21$). This indicates that call volume primarily reflects question difficulty rather than contributing directly to success.

### A.2.2 WHY ADDITIONAL TOOL CALLS DO NOT IMPROVE ACCURACY

Three observations explain why increasing call budgets does not yield higher scores:

**Tool-selection errors dominate.** Most failures arise from invoking the wrong tool or misinterpreting tool outputs. Additional calls often repeat the same pattern rather than correct it.

**Reasoning quality is the primary bottleneck.** Incorrect intermediate logic leads models to request irrelevant information or incorporate retrieved data incorrectly. More data does not correct upstream reasoning errors.

**No evidence of under-calling.** When a model knows which tool is relevant, it typically calls it immediately. High call counts reflect uncertainty rather than a lack of opportunity to call more tools.

### A.2.3 IMPLICATIONS

These findings indicate that CAIA performance is constrained by reasoning and tool-selection ability rather than the volume of external calls. Improvements in tool-augmented systems should therefore focus on better planning, tool routing, and evidence integration—not increasing call budgets.

### A.3 MODEL PERFORMANCE ACROSS 6 ANALYTICAL CATEGORIES OF TASKS

Tables 7 and 8 report majority-vote accuracy for all models across CAIA's six analytical categories. Together, they illustrate how model performance varies by category under both non–tool-enabled and tool-enabled settings.

### A.4 PROMPT TEMPLATES AND PROMPT VARIANTS

This section documents the core prompt used to elicit tool-augmented reasoning from models, together with a description of the prompt variants we experimented with during development. Our central design philosophy is that CAIA aims to test robust, default agent behavior rather than performance that depends on heavy manual prompt optimization. We experimented with several versions of the prompts to understand how different types of information affected model performance. In these prompts, we adjusted how much detail we provided about each tool (e.g., what the tool can do, what inputs it expects, and how the model should use it), and we adjusted the procedural guidance that explains how the model should combine the tools with its reasoning. The goal was to give enough structure for a fair comparison across models without adding so much instruction that the prompt itself became a major source of artificial performance gains. Arbitrary incremental improvements can always be manually added; therefore, prompt engineering is deliberately kept minimal. Importantly, even very simple tasks, such as querying a block timestamp or computing gas consumed on a given day, remain difficult for state-of-the-art models despite precise and direct prompts. This strongly suggests that poor performance is not due to lack of optimization, but instead inadequate grounding and tool-selection robustness.

### A.4.1 TOOL-ENABLED REASONING PROMPT

For models evaluated in the tool-enabled setting, we use a standardized reasoning prompt that frames the model as a crypto specialist with access to a fixed set of tools and requires it to decide iteratively whether to invoke tools or proceed to answer synthesis. The exact template is shown below.

```
You are a crypto specialist with access to comprehensive tools. Your job
    is to analyze questions and strategically select tools to gather the
    information needed to answer them.

Question:
{question}

You have access to the following tools:
{tools_intro}

You have gathered these tool results so far:
{tool_results_text}

Critical Instructions for Tool Selection:
- Analyze the question carefully to understand what information is needed
- SELECTIVELY choose tools based on the question's requirements
- Do NOT execute all available tools - be selective and strategic
- Consider what data each tool provides and how it relates to the
    question
- You can call multiple tools in a single iteration if they provide
    complementary information
- Review what information you have gathered so far from previous tool
    results (if any)

Agentic Workflow:
This is an iterative process. In each iteration, you must decide:
1. If you need more information: call the appropriate tools using the
    tool calling interface (up to 5 iterations total).
2. If you have enough information: do NOT call any tools. Provide
    reasoning and hand off to the synthesis step.

Output Format:
You MUST respond with a JSON object containing your decision and
    reasoning:
{
  "decision": "tool_call" or "synthesis",
  "reasoning": "Detailed reasoning explaining your decision, what
      information you have gathered so far, and why you're making this
      choice"
}

Decision Values:
"tool_call": Use this when you need to call tools to gather more
    information.
"synthesis": Use this when you have sufficient information and are ready
    to hand off to the synthesis step.

Reasoning Should Include:
- What information you have gathered so far.
- Why you are making this decision.
- If continuing tool calls: which tools and why.
- If synthesizing: why the gathered information is sufficient.

Decision Criteria:
- Review all tool results so far.
- Determine whether you have enough information.
- If yes: hand off to synthesis.
- If no: continue tool calls.
```

Table 7: Majority Vote Score (Not Tool-Enabled) across CAIA categories.

| Model | General Knowledge | On-Chain Analysis | Overlap | Project Discovery | Tokenomics | Trend Analysis |
|---|---|---|---|---|---|---|
| claude_4 | 1.000 | 0.104 | 0.071 | 0.102 | 0.000 | 0.000 |
| claude_opus_4 | 0.857 | 0.104 | 0.071 | 0.143 | 0.043 | 0.125 |
| claude_opus_4_1 | 1.000 | 0.104 | 0.071 | 0.122 | 0.087 | 0.000 |
| deepseek_r1 | 1.000 | 0.169 | 0.143 | 0.184 | 0.217 | 0.125 |
| deepseek_v3p1 | 0.857 | 0.143 | 0.214 | 0.163 | 0.000 | 0.000 |
| gemini_2.5_flash | 1.000 | 0.117 | 0.143 | 0.184 | 0.261 | 0.000 |
| gemini_2.5_pro | 1.000 | 0.156 | 0.143 | 0.245 | 0.261 | 0.125 |
| gpt_4.1 | 1.000 | 0.156 | 0.214 | 0.184 | 0.130 | 0.125 |
| gpt_4o | 1.000 | 0.130 | 0.143 | 0.163 | 0.087 | 0.125 |
| gpt_5 | 1.000 | 0.221 | 0.286 | 0.286 | 0.261 | 0.125 |
| gpt_o3 | 1.000 | 0.143 | 0.143 | 0.224 | 0.217 | 0.125 |
| gpt_oss_120b | 0.857 | 0.117 | 0.071 | 0.122 | 0.130 | 0.125 |
| grok_4 | 1.000 | 0.143 | 0.071 | 0.245 | 0.174 | 0.125 |
| grok_4_fast | 1.000 | 0.143 | 0.143 | 0.204 | 0.174 | 0.000 |
| kimi_k2 | 1.000 | 0.143 | 0.143 | 0.184 | 0.130 | 0.125 |
| llama_4 | 1.000 | 0.169 | 0.071 | 0.184 | 0.043 | 0.000 |
| qwen_3_235b | 1.000 | 0.130 | 0.143 | 0.143 | 0.130 | 0.000 |

Table 8: Majority Vote Score (Tool-Enabled) across CAIA categories.

| Model | General Knowledge | On-Chain Analysis | Overlap | Project Discovery | Tokenomics | Trend Analysis |
|---|---|---|---|---|---|---|
| claude_4 | 1.000 | 0.403 | 0.500 | 0.735 | 0.739 | 0.375 |
| claude_opus_4 | 1.000 | 0.403 | 0.571 | 0.755 | 0.739 | 0.250 |
| claude_opus_4_1 | 1.000 | 0.440 | 0.571 | 0.667 | 0.727 | 0.250 |
| deepseek_r1 | 0.571 | 0.208 | 0.143 | 0.061 | 0.261 | 0.000 |
| deepseek_v3p1 | 0.857 | 0.338 | 0.429 | 0.708 | 0.565 | 0.250 |
| gemini_2.5_flash | 0.857 | 0.299 | 0.357 | 0.449 | 0.435 | 0.000 |
| gemini_2.5_pro | 0.857 | 0.312 | 0.500 | 0.571 | 0.609 | 0.125 |
| gpt_4.1 | 1.000 | 0.325 | 0.500 | 0.592 | 0.565 | 0.250 |
| gpt_4o | 0.857 | 0.299 | 0.429 | 0.265 | 0.261 | 0.000 |
| gpt_5 | 1.000 | 0.571 | 0.786 | 0.735 | 0.783 | 0.500 |
| gpt_o3 | 1.000 | 0.442 | 0.714 | 0.571 | 0.652 | 0.125 |
| gpt_oss_120b | 1.000 | 0.545 | 0.571 | 0.714 | 0.783 | 0.250 |
| grok_4 | 1.000 | 0.364 | 0.429 | 0.551 | 0.435 | 0.250 |
| grok_4_fast | 1.000 | 0.468 | 0.571 | 0.735 | 0.826 | 0.375 |
| kimi_k2 | 0.857 | 0.286 | 0.643 | 0.673 | 0.522 | 0.250 |
| llama_4 | 0.714 | 0.234 | 0.286 | 0.184 | 0.261 | 0.125 |
| qwen_3_235b | 0.857 | 0.299 | 0.308 | 0.490 | 0.348 | 0.000 |

```
Analyze the question and the information you have gathered so far.
    Decide whether to continue with tool calls or hand off to synthesis.
```

### A.4.2 PROMPT VARIANTS EXPLORED

During development, we experimented with several prompt variants that modified:

- The level of detail provided in `tools_intro`, including how explicitly each tool's capabilities, expected inputs, and typical usage patterns were described.

- The amount of procedural guidance included in the narrative instructions (e.g., how strongly the prompt emphasized planning, tool selection, and evidence integration).

Across these variants, we observed that increasing specificity in tool descriptions or procedural hints could modestly shift tool-usage patterns, but did not eliminate the core failure modes identified in CAIA. Even under precise and direct prompting, models continued to fail on conceptually simple tasks such as block-timestamp retrieval or daily gas aggregation, reinforcing that the dominant bottlenecks lie in grounding and robust tool-selection rather than in prompt engineering alone.

## B  EXPERIMENT FRAMEWORK

### B.1  EXPERIMENT TOOLS

The functionality of the tools accessible to models during tool-augmented evaluation is listed below. The toolkit is intended to offer broad coverage of information sources pertinent to cryptocurrency and blockchain analysis. Where helpful, we include analogies to traditional financial research workflows to assist readers who may be less familiar with domain-specific terminology.

Table 9: Tool Catalogue Used in CAIA Evaluation

| Category | Tool Name | Description |
|---|---|---|
| Market Data | Price Feed | Lets the model see how a value changed over time, much like a basic line chart. |
| Market Data | Ranking Query | Highlights which assets are currently attracting the most attention, similar to a simple leaderboard. |
| Market Data | Positioning Analysis | Summarizes how traders are leaning (more optimistic or more defensive) using broad signals. |
| Market Data | Volatility Alerts | Flags recent market shake-ups so the model knows when volatility has spiked. |
| Technical Analysis | Indicator Analysis | Generates plain momentum/trend cues (e.g., "upward drift" vs. "cooling off") without requiring financial jargon. |
| Discovery/Search | Web Search | Finds public references related to a question and returns brief summaries. |
| Discovery/Search | URL Fetching | Pulls the text of an external article so the model can paraphrase or cite it directly. |
| Discovery/Search | Trend Detection | Points out topics or assets that are suddenly being discussed more than usual. |
| Social Sentiment | Twitter Search | Surfaces popular posts to show what people are currently talking about. |
| Social Sentiment | Social Stream | Provides curated discussions from recognizable commentators in one place. |
| DeFi Metrics | Protocol Metrics | Offers a broad sense of whether a project or application is seeing more or less engagement. |
| DeFi Metrics | Revenue Analysis | Gives a lightweight indication of whether people are paying to use a service. |
| DeFi Metrics | Network Activity | Summarizes overall activity on a larger network so relative interest can be compared. |
| DeFi Metrics | Exchange Summary | Shows where trading or usage is concentrated across platforms. |
| DeFi Metrics | Exchange Overview | Rolls multiple signals into a "big picture" summary for quick orientation. |
| On-Chain Data | Supply Query | Answers "how much of this asset exists" in plain terms. |
| On-Chain Data | Data Query | Reads publicly available ledger data so the model can confirm basic facts. |
| On-Chain Data | Time Conversion | Converts a calendar time into the ledger's internal reference, making it easier to find past events. |
| On-Chain Data | Balance Query | Reports what a particular account currently holds. |
| On-Chain Data | Transaction Query | Shows what happened in a specific ledger entry. |
| On-Chain Data | Block Query | Gives the headline information for a single ledger page (time, order, etc.). |
| Token Analytics | Release Schedule | Lists upcoming moments when more of an asset becomes available, similar to a basic release calendar. |
| Execution | Code Execution | A small workspace for quick arithmetic or comparisons when needed. |

## C  ILLUSTRATIVE TASK EXAMPLES

Each CAIA item consists of a question, its expert-verified ground-truth answer, and a category label. The examples below correspond to tasks highlighted in the rebuttal and demonstrate how even simple, deterministic questions expose systematic weaknesses in model tool selection and reasoning.

### C.1  EXAMPLE 1: COUNTING UNIQUE ERC–721 TRANSFER PARTICIPANTS

**Question.**  *How many unique wallets had at least one ERC–721 transfer of Bored Ape Yacht Club tokens during calendar Q4 2024?*

**Ground-Truth Answer.**  `1,966.00`

**Category.**   **On-chain analysis**

**Walkthrough.**   The task requires enumerating all ERC–721 transfer events for the Bored Ape Yacht Club collection within the Q4 2024 date window and counting distinct participating wallets. The answer is objectively recoverable through deterministic on-chain data.

**Observed Failure Modes.**

- relying on generic Web2 search instead of deterministic on-chain tools,
- extracting outdated or fabricated values from SEO–optimized pages,
- misinterpreting summaries or dashboards instead of using verifiable event data,
- returning total transfers rather than unique wallets.

### C.2   Example 2: Retrieving a Block Timestamp

**Question.**   *Retrieve the block timestamp for Ethereum block #19,560,000.*

**Ground-Truth Answer.**   A precise, verifiable UNIX timestamp corresponding to the block's inclusion time (omitted here for formatting consistency; canonical values are obtainable through standard block explorers).

**Category.**   **On-chain analysis**

**Walkthrough.**   Although trivial via any authoritative block explorer or metadata query, this task surfaces common routing errors: the timestamp is deterministic, yet many models fail to retrieve it correctly.

**Observed Failure Modes.**

- routing the query through general Web2 search instead of authoritative blockchain tools,
- using outdated or misleading search snippets,
- hallucinating improvised formulas to approximate timestamps rather than retrieving the deterministic value.

### C.3   Discussion

These examples demonstrate that CAIA tasks are simple only when models correctly identify and use authoritative tools. Failures stem primarily from incorrect source selection rather than computational difficulty. The resulting behaviors reveal structural fragility in reasoning and grounding that persists even for objectively straightforward queries.

## D   CAIA Task Categories: Definitions and Representative Examples

To improve clarity and interpretability of CAIA's task design, we provide formal definitions of the six analytical categories used throughout the benchmark, together with representative examples drawn from the curated dataset. These definitions expand upon Table 1 in the main paper and illustrate the specific competencies assessed in each category.

### D.1   On-Chain Analysis

On-Chain Analysis tasks evaluate an agent's ability to extract, interpret, and validate immutable blockchain ledger data. These items require precise interaction with block-level records, historical

state queries, or timestamp-based lookups. Successful solutions often depend on correct use of authoritative on-chain data sources (e.g., block explorers, node APIs) and accurate reasoning about ledger structure.

**Representative Examples.**

- Retrieve the block timestamp for Ethereum block 19,560,000.
- Query USDC's total supply on Ethereum at block 19,000,000.

D.2 PROJECT DISCOVERY

Project Discovery tasks assess an agent's ability to identify official project information, verify canonical contract addresses, and differentiate authoritative sources from misleading or unofficial ones. These tasks stress-test source selection in environments where spoofed websites, outdated documentation, and SEO-driven misinformation are common.

**Representative Examples.**

- What is the Uniswap V3 router contract address on Ethereum mainnet?
- What's Pendle's current router contract address?

D.3 TOKENOMICS

Tokenomics tasks measure understanding of supply mechanics, vesting and unlock schedules, and circulating-supply calculations. These problems combine economic reasoning with contract-level data and temporal constraints, often requiring multi-step aggregation or careful interpretation of release schedules.

**Representative Examples.**

- How many MYSHELL tokens unlock between 2025-06-30 and 2025-09-30?
- What is the circulating supply of `TRUMP` as of 2025-08-30?

D.4 OVERLAP

Overlap tasks require comparative or relational reasoning across multiple protocols, chains, or assets. They frequently involve multi-source synthesis, ratio or share calculations, or juxtaposition of parallel ecosystem metrics. These items probe an agent's ability to coordinate disparate information streams and maintain consistency across them.

**Representative Examples.**

- What share of total L2 TVL does Base represent on 2025-08-30?
- Which had higher TVL on 2025-09-05: HypurrFi or Hyperlend?

D.5 TREND ANALYSIS

Trend Analysis tasks examine an agent's ability to interpret temporal patterns in on-chain or market activity. These tasks may require counting events across time windows, comparing historical periods, or relating observed shifts to broader protocol or market events. Solutions typically depend on correct extraction of time-series data and coherent temporal reasoning.

**Representative Examples.**

- How many unique tokens were launched on Pump.fun in August 2025 compared to January 2025?
- What major event caused the Cyber Token price surge in August 2025?

### D.6 GENERAL KNOWLEDGE

General Knowledge tasks assess conceptual understanding of blockchain mechanisms, cryptographic primitives, protocol architectures, and economic design principles. These items emphasize explanatory synthesis rather than retrieval, requiring agents to integrate background knowledge into clear, coherent, and technically correct summaries.

**Representative Examples.**

- What is Optimism and the Superchain? Explain how chains are connected.
- What keeps the Bitcoin network running once all blocks have been mined?

## E ADVERSARIAL THREAT MODEL AND INFORMATION ENVIRONMENT

Although many CAIA tasks appear conceptually simple, the surrounding information environment is adversarial by default. Models are not directly connected to blockchain consensus layers; instead, they operate across heterogeneous sources including Web2 search, user-generated content, social channels, and domain-specific analytics. This creates systematic opportunities for manipulation and misinformation.

### E.1 THREAT MODEL

CAIA evaluates an agent's ability to select authoritative sources and avoid adversarial or unreliable ones. We define adversarial pressure along three concrete axes:

**1. SEO-Driven Misinformation.** Crypto markets are routinely targeted by search-engine manipulation, artificial amplification of misleading content, and spam domains designed to capture users searching for token tickers or protocol names. These pages often embed outdated or fabricated statistics (TVL, supply, or addresses).

**2. Spoofed or Clone Websites.** Attackers frequently deploy visually indistinguishable clones of protocol websites with altered contract addresses. A single incorrect contract address can lead to irreversible monetary loss in real settings. Several model traces demonstrated susceptibility to such spoofed domains when relying on Web2 search.

**3. Social Media Manipulation.** Platforms such as X/Twitter and Reddit contain a high volume of strategically misleading posts, wash-traded narrative amplification, and misinformation campaigns surrounding token launches, exploit events, and governance votes.

### E.2 REAL-WORLD MOTIVATING INCIDENTS

To ground the threat model, we highlight two publicly documented incidents demonstrating that misinformation is not hypothetical but endemic:

- **SEC X-Account Compromise (2024).** A false post announcing Bitcoin ETF approval led to significant market volatility before the breach was corrected.
- **Fake Walmart–Litecoin Partnership (2021).** A fabricated press release, widely circulated by reputable news aggregators, briefly moved markets before being debunked.

Such incidents illustrate the difficulty of source verification when information is rapidly disseminated and economically consequential.

### E.3 HOW ADVERSARIAL NOISE AFFECTS AGENT BEHAVIOR

During CAIA evaluation, we observed models:

- prioritizing high-ranking Web2 search snippets over authoritative blockchain tools;

- selecting outdated or inconsistent values from SEO-optimized blogs;

- hallucinating contract addresses or token supplies based on partial textual cues;

- failing to cross-check conflicting results across tools.

These behaviors occur even though correct answers are deterministically recoverable through the provided tool suite. The adversarial environment therefore amplifies inherent weaknesses in tool selection and reasoning.

### E.4 WHY CRYPTO PROVIDES A UNIQUE TESTBED

The cryptocurrency ecosystem possesses a combination of properties not found together in other evaluation domains:

- **Deterministic, immutable ground truth:** blockchains provide transparent, verifiable on-chain data.

- **High financial stakes:** minor analytical mistakes may imply irreversible loss.

- **Strong incentives for manipulation:** misinformation campaigns are economically motivated.

- **Temporal contamination resistance:** tasks tied to recent state (block heights, unlock dates) minimize pretraining leakage.

This makes crypto an ideal environment for testing not only correctness but *resilience*: the ability of agents to avoid being misled when multiple plausible but contradictory information sources are present.

## F HUMAN ANALYST BASELINE

### F.1 PARTICIPANT RECRUITMENT AND SCREENING

To establish a human reference point for the benchmark, we recruited individuals with realistic and heterogeneous crypto related experience. Participants were drawn from university blockchain organizations and early career research or investment teams, representing typical profiles of practitioners who routinely engage with crypto analytical tasks.

Each participant completed a short diagnostic assessment covering tokenomics, project due diligence, and basic on-chain mechanics. Individuals who did not meet a minimum competency threshold were excluded to avoid downward bias in the baseline estimate.

### F.2 EVALUATION PROTOCOL

Qualified participants were assigned a randomized subset of benchmark questions spanning all categories and difficulty levels. To ensure methodological parity with model evaluation, participants were restricted to the same publicly accessible tools and information sources available to the LLMs. External materials, collaboration, or online investigation beyond these standardized resources were not permitted.

Responses were evaluated against expert verified ground truth answers. Credit was assigned only for exact matches or semantically equivalent formulations explicitly approved by domain experts.

### F.3 PERFORMANCE CHARACTERISTICS

We observed substantial variation across participants. Those with stronger technical backgrounds tended to perform well on protocol design and implementation questions but occasionally missed domain specific documentation details. Conversely, participants with operational or investment-focused backgrounds demonstrated higher accuracy on tokenomics and due-diligence items but showed difficulty with lower-level on-chain reasoning.

Per individual accuracy was computed, and the highest and lowest performers were removed as statistical outliers. The human baseline was defined as the mean accuracy of the remaining participants. This baseline provides a stable reference for analyzing gaps between human reasoning and model behavior. Future iterations of the benchmark will include finer-grained comparisons across question types, difficulty levels, and skill profiles.

# G  DATASET CONSTRUCTION AND CURATION PIPELINE

## G.1  CROWDSOURCING AND INITIAL COLLECTION

The benchmark dataset was constructed through a multi-stage pipeline designed to capture realistic informational demands within the crypto ecosystem. We conducted an open call for question submissions across public crypto communities (primarily on Twitter/X) and compensated contributors for high quality, domain relevant submissions. Off topic, speculative, or unverifiable questions were excluded.

To complement crowdsourced inputs, experienced builders and contributors from established protocols were invited to submit questions reflecting analytical challenges encountered in professional contexts.

## G.2  AUTOMATED PREPROCESSING

Given the heterogeneity of the raw submissions, we employed a large language model as an initial preprocessing tool. The model standardized formatting, flagged near duplicate questions, and identified ambiguous, unanswerable, or unverifiable prompts. For instance, short-term price predictions or context deficient generic questions were automatically filtered out. All automated preprocessing decisions were manually reviewed by domain experts to ensure correctness.

## G.3  EXPERT REVIEW AND ANNOTATION

After preprocessing, domain experts—including protocol engineers, research analysts, and security specialists—conducted a comprehensive review of the remaining questions. Each question was evaluated for clarity, factual answerability, and relevance. Experts also assigned metadata such as topic category (e.g., governance, tokenomics, protocol design, on-chain analysis), expected difficulty, and required domain knowledge.

When needed, questions were rephrased to reduce ambiguity or to normalize terminology while preserving the intended meaning.

## G.4  GROUND TRUTH AND DATASET FINALIZATION

Following iterative refinement, the pipeline yielded 178 curated questions paired with canonical ground-truth answers. Each answer was validated using authoritative sources, including protocol documentation, audited codebases, and on-chain analytics dashboards. When multiple semantically equivalent formulations existed, they were consolidated into a single normalized answer to ensure consistent scoring across human and model evaluations.

Questions were then standardized into a uniform format with clearly defined scope, consistent naming conventions, and self-contained phrasing. The resulting dataset provides broad coverage of real world information seeking behaviors in the crypto ecosystem.

# H  ETHICAL CONSIDERATIONS

All human participants were compensated at rates consistent with standard academic user study practices, using prepaid gift cards. Compensation was independent of performance and provided regardless of task outcome.

Participants were informed of study procedures, approximate time commitment, and their right to withdraw at any time. No personally identifiable information beyond basic demographic and

expertise-level self reports was collected. All tasks were completed individually, under non-coercive conditions, and using only standardized resources.

The study involved minimal-risk activities and adhered to institutional guidelines for human subjects research. Recruitment, screening, compensation, and participant protections are documented explicitly in the revised manuscript.

