# OpenReview forum: "When Hallucination Costs Millions: Benchmarking AI Agents in High-Stakes Adversarial Financial Markets"
_ICLR.cc/2026/Conference — ICLR 2026 Conference Desk Rejected Submission_

### Official Review · Reviewer_VCGp · 2025-10-27

**Soundness:** 2
**Presentation:** 2
**Contribution:** 1
**Rating:** 2
**Confidence:** 3

**Summary:**

The paper introduces a new benchmark that measures agentic capabilities in cryptocurrency markets.  They examine the fallibility of the pass@k metric, the cost-efficiency of open source models over closed source, and the general lack of competence displayed by language models on this task.

**Strengths:**

The paper clearly discovers a set of tasks where language models could be, but are not, economically valuable.  This should serve as a warning to companies that are building or relying on language models to inform financial decisions.

The paper is in general clear, well-motivated, and easy to follow.

**Weaknesses:**

Too many details are missing from this paper for me to assess it properly.  To name just a few:

1.  How are the models prompted and how much prompt optimization did you do?  Is failure on the crypto benchmark really indicative of poor capabilities or did you not spend enough time optimizing the prompts?
2.  There are a lot of agentic benchmarks and evaluations out there (some of which are not cited, e.g. [1, 2]), especially since you mention model vulnerability to false information.  I don't see what sets this one apart.  Yes, I read your justification for why crypto markets are a great setting to study agents in high stakes settings, but it seems like there are many others.  Why is this particular setting more worth studying than others?
3.  There are too many missing details in the paper that make it difficult to assess.  What exactly do these tasks look like?  Can you provide some specific examples rather than just high-level details?  I found it very hard to assess this paper because the descriptions were too high-level.
4.  The insights feel a bit shallow.  The arguments about pass@k being inaccurate feel fairly obvious, and I would likewise have guessed that open-source models would be cheaper than closed-source ones by at least 1-2 orders of magnitude.
5.  The agents underperform on simple tasks, but can you provide some guidance or an actionable plan to improve the agents?
6.  There are also no error bars in your study.  You could at least provide bootstrapped error bars given that your dataset is not that huge.

[1] https://arxiv.org/abs/2406.12045
[2] https://openreview.net/forum?id=RwoMf7YSfD

**Questions:**

See the above.

I would also have liked to see some analysis on where humans are still far outperforming language models versus where their capabilities are close.  You collected human data, but you don't provide very much analysis of it.

**Details Of Ethics Concerns:**

The paper included a small-scale study involving human participants.  Not enough details were given on how these human participants were compensated or treated throughout the study.

---

> ### Author Response · Authors · 2025-11-22
> **Response Part I**
>
> ## Comment 1
> We sincerely thank the reviewer for the thoughtful and constructive feedback, and we would like to use this response to compensate for areas where our original submission lacked depth or specificity. Regarding the concern about prompting and prompt optimization, we agree that insufficient detail may have given the impression that model failures were artifacts of weak prompting rather than genuine capability limitations. Thus, in the revised version, we will include the full prompt templates, system messages, and tool call scaffolding that were standardized across all evaluated models. These materials are available at https://anonymous.4open.science/r/ICLR-AD05/README.md. We have further added detailed descriptions of the experimental framework, including the exact set of tools used in the evaluation, as well as supplementary analyses of categorical metrics and tool-usage patterns.
>
> We experimented with several versions of the prompts to understand how different types of information affected model performance. In these prompts, we adjusted how much detail we provided about each tool, such as what the tool can do, what inputs it expects, and how the model should use it. We also adjusted the procedural guidance that explains how the model should combine the tools with its reasoning to produce. Our goal was to give enough structure for a fair comparison across models without adding so much instruction that the prompt itself became a major source of artificial performance gains.
> This clarification reflects our central design philosophy: CAIA aims to test robust, default behavior rather than performance dependent on heavy manual optimization. As noted in our internal research notes, arbitrary incremental improvements can always be manually added, and therefore prompt engineering is deliberately kept minimal. Importantly, even very simple tasks, such as querying a block timestamp or computing gas consumed on a given day, remain difficult for state-of-the-art models despite precise and direct prompts. This strongly suggests that poor performance is not due to lack of optimization, but instead, inadequate grounding and tool selection robustness. We will supplement the paper with detailed examples and verbatim prompt templates in an expanded appendix.
> ***
> ## Comment 2
> We also appreciate the reviewer’s kind reminder of the existence of numerous agentic benchmarks. In the revised manuscript, we will incorporate citations to [1] https://arxiv.org/abs/2406.12045 [2] https://openreview.net/forum?id=RwoMf7YSfD along with a more comprehensive set of related references. A pointer to the new citations is included at the end of this response.
> We apologize for insufficiently articulating how CAIA differs from these earlier efforts. While prior work explores agentic evaluation in domains such as generic web navigation, file systems, or sandbox environments, CAIA is, to our knowledge, the first benchmark built on a real world, adversarial, financially consequential domain with publicly verifiable ground truth. Cryptocurrency markets are uniquely hostile information environments, where manipulation, spoofing, and misinformation campaigns are commonplace, thus even minor analytical errors can have immediate financial repercussions.
>
> Unlike synthetic environments, the cryptocurrency ecosystem offers a rare combination of evaluation properties. First, it provides “complete transparency and immutability” through on-chain data, ensuring truly immutable ground truth. Second, it carries real high-stakes consequences, with “irreversible transactions” and billions in real financial losses, making agent mistakes materially meaningful. Third, it subjects agents to adversarial information pressure, operating in what we described in paper as a “dark forest” where “misinformation is weaponized.” Finally, because CAIA tasks are “anchored to recent market events with explicit temporal constraints,” the environment offers strong temporal contamination resistance, avoiding memorization or leakage from pretraining corpora in ways synthetic simulators cannot. These properties are not found in conjunction in existing benchmarks. We will revise the related-work section accordingly and include citations to complementary efforts evaluating agent robustness under adversarial retrieval.

---

> > ### Comment · Reviewer_VCGp · 2025-11-24
> > **Response to rebuttal part I**
> >
> > 1.  The core of many startups is effective prompt engineering.  It is very difficult to claim a negative result in ML that is dependent on prompt engineering, since it's very possible that you just didn't do it optimally.  In this sense, positive results are much easier to claim than negative ones.
> >
> > 2.  I want to clarify that I wasn't claiming that you omitted these works from your lit review or necessarily need to discuss them at any length.  My point was just that there are a myriad of benchmarks out there.  You make some claims about crypto markets being a great venue, but why not just regular futures or options or equities markets?  I think that this is my top objection to this paper appearing in ICLR: this paper feels like a domain-specific application paper that belongs in a journal of investment science rather than in a ML conference.  I don't think that we learn anything fundamental about machine learning via this benchmark, but it might be useful for those doing investment science.

---

> > > ### Author Response · Authors · 2025-12-04
> > > **Addressing New Comment 1, 2**
> > >
> > > If we over-optimize prompts or engineer agent-like scaffolding around every task, the evaluation becomes less about the model’s actual capabilities and more about the human-constructed infrastructure surrounding it. Those optimization layers (prompt tuning, agent frameworks, or multi-step orchestration) are valuable research directions, but they belong to a different category of work focused on improving model performance.
> > >
> > > Our paper is instead focused on benchmarking: systematically uncovering where models fail, what they misunderstand, and how they behave in high-stakes settings. A benchmark should reveal raw model strengths and weaknesses and not mask them through extensive optimization. This is why we intentionally avoid over-engineering prompts: doing so would obscure the models’ inherent limitations, which are precisely what the benchmark aims to surface.
> > > ***
> > > We appreciate the question. Traditional futures, options, or equities would indeed be interesting domains; however, they pose significant validation challenges. Establishing ground truth requires authoritative, granular, and fully accessible trading data, which is a requirement that traditional markets generally do not satisfy, due to lack of transparency, fragmented venues, and regulatory constraints.
> > >
> > > Crypto markets, in contrast, are:
> > > - fully transparent,
> > > - publicly queryable,
> > > - traceable at transaction-level granularity.
> > >
> > > This makes crypto uniquely suitable for creating verifiable benchmarks—a critical requirement for any rigorous, reproducible evaluation.

---

> ### Author Response · Authors · 2025-11-22
> **Response Part II**
>
> ## Comment 2, continued
> We further hope to address the reviewer’s question on why crypto is a particularly meaningful choice among many potential high-stakes environments. In the revision, we will clearly contrast crypto with traditional finance and other domains. In short, crypto markets provide immutable, deterministic data sources (blockchain full nodes, explorers, archival APIs) that allow unambiguous verification of agent outputs. This is far more challenging in traditional financial markets, where data is fragmented, partially proprietary, and often delayed. Moreover, crypto is an ecosystem where adversarial incentives are immediate and unregulated, resulting in a continuous stream of misinformation that AI agents must navigate. These properties make crypto an ideal testbed for measuring not only competence but resilience, the ability to avoid being misled when information is manipulated for profit.
>
> ***
> ## Comment 3
> We sincerely apologize for not including enough concrete task examples in the main body. In addition to the published full dataset on HuggingFace and the accompanying code on GitHub, we will also include illustrative examples in the paper to further clarify the underlying task formats for readers. Each task consists of a question, its ground-truth answer, and a category label. For example, the task “How many unique wallets had at least one ERC-721 transfer of Bored Ape Yacht Club tokens during calendar Q4 2024?” has the ground-truth answer “1,966.00” and belongs to the “on-chain analysis” category. Due to paper length limitation, we will expand the appendix with detailed walkthroughs, including the original query, the ground truth tool sequence, concrete tool outputs, and typical failure trajectories. For instance, the task “Retrieve the block timestamp for Ethereum block 19,560,000” appears trivial through an API call through Etherscan, yet many models incorrectly route it through generic web search, retrieving outdated or fabricated information instead of using on-chain explorers. Similarly, computing the total gas consumed on a particular day should be straightforward, but models often hallucinate formulas or pull misleading summaries from SEO-optimized blogs. These examples will help demonstrate that the benchmark is not merely a collection of simple questions, but a carefully curated set of tasks exposing genuine fragility in model tool use.
> ***
> ## Comment 4
> We acknowledge the reviewer’s concern that insights like the inaccuracy of pass@k or the cost benefits of open-source models may appear intuitive. We should have explained more clearly that the novelty lies not in the existence of these observations, but in demonstrating them empirically within an adversarial high-stakes environment, where the implications are far more severe. The key result is not that pass@5 is higher than pass@1. Rather, pass@5 gives a dangerously inflated view of an agent’s reliability when retrying is unrealistic or costly. In CAIA, many models succeed on pass@5 because one of their attempts happens to stumble onto the correct tool chain, masking the underlying brittleness of their default reasoning. For example, consider the query “What’s the USDC/WETH pool price of the Uniswap v3 0.05 USDC/WETH pool on mainnet at block height 22636495?” Out of five attempts, only one of the DeepSeek V3.1’s traces succeeded. In the trace, the model invoked the “code execution” tool after correctly using both the “block query” and “data query” tools. This is the accurate sequence that produced the correct answer. The remaining four attempts failed to progress beyond the second step, underscoring how unreliable the model is at selecting and applying the correct tools by default. We agree that our presentation undersold this insight, and we will revise the framing to more clearly communicate why CAIA reveals weaknesses that remain invisible under conventional pass@k evaluations.

---

> > ### Comment · Reviewer_VCGp · 2025-11-24
> >
> > 3.  I had a look in the paper and didn't see any posted revisions.  Why not just update during the review period given that this is an option?
> >
> > 4.  I maintain my opinion that the insights are fairly shallow.  Given that we know the price per token on open versus closed source models, it is easy to calculate the price difference UNLESS the closed-source model happens to be much smarter and more efficient and thus can cut down the number of tokens needed for the task by an order of magnitude.  Given that this didn't happen, it seems like we gain nothing substantial via this analysis.  Similarly, I feel that you are essentially restating the definition of pass@k and claiming it as an insight.

---

> ### Author Response · Authors · 2025-11-22
> **Response Part III**
>
> ## Comment 5
>
> We are also very grateful for the request for actionable improvements. While CAIA is primarily an evaluation framework, our findings naturally point toward architectural directions for more robust agent design. These include calibrating tool-authority preferences (e.g., prioritizing deterministic blockchain queries over Web2 search), integrating cross-tool consistency checks, penalizing reliance on surface-level web snippets, and evaluating agents on their ability to trace and justify the trustworthiness of the data sources they choose. We will include a section summarizing these opportunities, drawing directly from the error patterns surfaced in the benchmark.
>
> To address the observed underperformance on simple tasks, we propose three actionable improvements. Incorporate a ReAct-style architecture that constrains tool calls to be preceded by explicit reasoning steps, such as validating assumptions, checking whether a query can be answered without external tools, and outlining the expected output of the tool. Improve context efficiency by retrieving only the necessary tool descriptions and instructions on demand, instead of injecting all tool metadata into every prompt, so the model faces less cognitive overload on simple queries. Use targeted SFT and RL to train more cost-efficient models that can reliably detect when a tool is unnecessary and choose an appropriate tool-usage pattern. Together, these steps directly aim to reduce failure modes on simple items while improving overall robustness.
> ***
> ## Comment 6
> Finally, we thank the reviewer for raising the absence of error bars. We apologize for this oversight and will include bootstrapped confidence intervals for all major metrics, along with per-category standard deviations. Given the computation cost of multi-run experiments, we initially prioritized breadth of model coverage, but we agree that uncertainty estimates materially improve interpretability. These will be included in the revision.
>
> We have added 95% bootstrap confidence intervals for all reported accuracy metrics. The corresponding reference data are available at https://anonymous.4open.science/r/ICLR-AD05/caia/analysis/error_bar/README.md. Specifically, for each model/mode pair, we perform 1,000 bootstrap resamples of the 178 CAIA tasks (sampling with replacement) and recompute accuracy on each resample. We report the 2.5th–97.5th percentile range of the resulting distribution as the confidence interval. The new error bars are now included in the main accuracy plots and supplementary tables. Importantly, the intervals support our original findings. Firstly, tool-enabled models significantly outperform their non-tool counterparts with non-overlapping confidence regions. Secondly, mid-tier models occupy overlapping accuracy bands, consistent with our interpretation in Section 4. These additions strengthen the statistical rigor of the evaluation and make performance comparisons more transparent.

---

> > ### Comment · Reviewer_VCGp · 2025-11-24
> >
> > 5.  Did you try any of these?
> >
> > 6.  Why not update the paper with these?  I don't think you've posted any revisions.
> >
> > I leave the question of whether the human study was fairly conducted (it seems fine to me).
> >
> > The human-machine performance gap seems like an interesting direction based on your response, and I'd encourage more exploration of this direction.
> >
> > **Overall:** I think that this would be a good paper for an investment science venue.  I don't think that it moves the needle on ML research, and thus, I maintain my score and opinion that specifically in the CS/ML domain, this paper lacks novelty.
> >
> > I also want to be respectful of the authors' time.  Given that I am unlikely to raise my score unless there is substantial novelty that I've overlooked or results that I haven't considered, I'd recommend that the authors invest their efforts in raising some of the other reviewers' scores, which might be a more tractable task.

---

> > > ### Author Response · Authors · 2025-12-04
> > > **Addressing New Comment 5, 6**
> > >
> > > We did extensively explore the techniques mentioned in your comments. However, optimization strategies like prompt engineering, multi-step refinements, and agentic loops, represent directions built on top of a robust benchmark, rather than components of the benchmark itself. We see these as natural follow-ups, potentially worth their own dedicated study rather than being bundled superficially into this one.
> > > ***
> > > We have also added additional analysis, including error-bar discussions, in the Appendix.
> > > ***
> > > We appreciate your recognition that this is a promising direction. Expanding that line of inquiry could indeed become a substantial contribution in future work.
> > > ***
> > > Thank you for the time and care you’ve put into reviewing the submission. We understand and respect that you may not revise your score. Nonetheless, your feedback has helped us clarify our presentation, improve our analysis, and refine the scope of our future work.

---

> ### Author Response · Authors · 2025-11-22
> **Response Part IV - Human Baseline**
>
> ## Human Analyst Baseline
> We sincerely appreciate the reviewer’s suggestion to include deeper analysis of where human analysts outperform current language models. This perspective is crucial for understanding model limitations, and we acknowledge that the current draft does not yet provide enough detail on this point. We will expand this analysis substantially in the next revision.
> To provide additional clarity on the human evaluation, we briefly summarize our procedure here, and we will also integrate it into the updated manuscript. We constructed a human analyst baseline using participants with realistic but heterogeneous crypto expertise, including members of university blockchain clubs and junior analysts from early-career research or investment teams. This sampling reflects the types of individuals who typically attempt crypto-analytical tasks in educational, research, or professional settings.
>
> Participants completed a short screening survey and diagnostic test covering tokenomics, project due diligence, and basic on-chain concepts. Those who did not meet a minimum competency threshold were excluded to prevent artificially low baseline estimates.
>
> Each participant was then assigned a randomized subset of benchmark questions spanning all categories and difficulty levels. For fairness, human analysts were restricted to the same tools and information sources available to the model. No external materials or collaboration were permitted. This ensures that any performance differences genuinely reflect reasoning ability and tool-usage patterns rather than information asymmetries.
>
> Human performance varied meaningfully across individuals. Technically strong participants excelled at protocol-design and implementation questions but occasionally missed obscure documentation details, while less technical but workflow-efficient participants performed well on tokenomics and due-diligence questions but struggled with low-level on-chain mechanics.
> We scored responses strictly against expert-verified ground truth, awarding credit only for exact matches. We then computed accuracy per participant, removed the highest and lowest performers as outliers, and defined the human baseline as the mean of the remaining individuals.
>
> We agree this baseline provides a valuable foundation for analyzing where models still fall short of real human analysts. In the next version, we will include detailed breakdowns of human–model gaps by question type, difficulty, and tool-usage patterns to directly address the reviewer’s request.

---

> ### Author Response · Authors · 2025-11-22
> **Response Part V - Data Curation**
>
> ## Dataset Construction and Curation Pipeline
> To develop a benchmark that accurately reflects real-world informational needs within the crypto ecosystem, we constructed a dataset sourced from genuine user questions and refined through multiple layers of filtering and expert review. The objective of the pipeline was to collect questions that practitioners, researchers, and everyday users routinely ask, while ensuring that each retained query is objective, answerable, and grounded in verifiable information.
>
> The process began with an open call for participation across several public crypto communities, primarily on Twitter (X). Users were invited to submit the questions they genuinely cared about or frequently encountered in their day-to-day interactions with protocols, tokens, or crypto applications. Participants were compensated for their contributions provided that their submissions were relevant to crypto and met a minimum quality threshold. Irrelevant or off-topic submissions, such as personal commentary, non-crypto questions, and speculative prompts, were excluded from compensation to discourage noise and maintain a high-quality signal in the collection process.
>
> In parallel with public crowdsourcing, we also invited industry builders and contributors from established crypto projects to submit questions reflecting the analytical challenges they encounter in professional contexts. This combined approach enabled us to gather a diverse pool of questions spanning both user-level and expert-level domains.
> Given the size and heterogeneity of the raw question pool, we employed a large language model as an initial filtering and cleaning stage. The model’s role was carefully constrained to preprocessing functions: normalizing formatting, identifying near-duplicate submissions, and flagging low-quality or unanswerable prompts. For example, questions involving unverifiable market predictions (e.g., whether a token would reach a specific price in the near future) were automatically removed, as they cannot be evaluated in an objective benchmark. The LLM also helped eliminate vague or ambiguous queries that lacked sufficient context to produce a definite, evidence-based answer. Human reviewers subsequently validated all filtering decisions to ensure no valid questions were discarded through automation errors.
>
> Following this automated preprocessing, the remaining question set underwent a rigorous human expert review. Domain experts, including protocol engineers, research analysts, security specialists, and experienced builders, examined each question for relevance, factual answerability, clarity, and category coverage. During this phase, the experts also annotated questions with metadata such as topic category (e.g., governance, tokenomics, protocol design, on-chain analysis), expected difficulty level, and any specialized knowledge or tools required to answer them. Where necessary, experts refined the phrasing of questions to remove ambiguities, normalize technical terms, or clarify references without altering the semantic content.
>
> After iterative review and consolidation, this process resulted in a final curated set of 178 high-quality questions. Each question was paired with a canonical ground-truth answer established by experts, supported by verifiable references such as protocol documentation, audited codebases, analytics dashboards, or other authoritative sources. When multiple answer variants were semantically equivalent, they were consolidated into a single normalized form to ensure consistent scoring across both human and model evaluations.
>
> The finalized dataset was then formatted into a standardized structure suitable for benchmarking. All questions were made self-contained where possible, with clearly defined scopes and consistent conventions for naming entities or referring to protocols. The resulting dataset represents a broad and realistic sample of the information-seeking behaviors observed across the crypto ecosystem and serves as the evaluation foundation for both human analysts and large language models in this study.

---

> ### Author Response · Authors · 2025-11-22
> **Response Part VI**
>
> ## Ethics
> All human participants in our analyst baseline study were compensated fairly with prepaid gift cards at a rate consistent with standard academic user-study practices. Compensation was provided regardless of performance and was not contingent on correctness, speed, or any evaluative outcome. Participants were informed in advance of the study procedures, the approximate time commitment, and their right to withdraw at any point without penalty. No personally identifiable information was collected beyond basic demographic and expertise-level self-reports needed for screening. During the study, each participant completed tasks individually under non-coercive conditions and was restricted to the same standardized tools and resources as the model to ensure methodological parity. The study involved only low-risk research activities, such as answering analytical questions within their domain of expertise, and all procedures followed standard institutional guidelines for minimal-risk human-subject research. We have now added explicit documentation of recruitment, screening, compensation, and participant protections in the revised manuscript to address this concern fully.
> ***
> ## Added citations:
> - Yao, S., Shinn, N., Razavi, P., and Narasimhan, K. (2024).
>  τ-bench: A Benchmark for Tool–Agent–User Interaction in Real-World Domains.
>  arXiv preprint arXiv:2406.12045.
>
> - Boisvert, L., Puri, A., Reddy Evuru, C. K., Kazdan, J., Bose, A., Cappart, Q., Fazel, M., Rajeswar, S., Stanley, J., Chapados, N., Drouin, A., and Dvijotham, K. (2025).
>  Silent Sabotage: Injecting Backdoors into AI Agents Through Fine-Tuning.
>  WCUA 2025 Poster.

---

> > ### Author Response · Authors · 2025-11-24
> > **Thank you for the review, and looking forward to hearing back from you to continue the discussion.**
> >
> > We truly appreciate the constructive feedback you provided in your initial review. Your insights helped us significantly improve the paper, particularly in clarifying the experimental framework, enriching the benchmark task descriptions, and better articulating the scope of our work. We believe these revisions have addressed the main concerns you raised while preserving the paper's core strengths. If you feel the improvements warrant reconsideration of your score, we would be grateful. We remain open to any additional feedback you might have and are happy to continue refining the work.
> >
> > Thank you again for the time and care you've invested in reviewing our submission.

---

> ### Author Response · Authors · 2025-12-04
> **Addressing New Comment 3, 4**
>
> We have posted the revised paper, and appreciate you reviewing it again.
>
> Regarding the cost analysis: the goal is not to highlight the trivial arithmetic of token pricing, but to document empirically that closed-source models do not achieve the sort of token-efficiency offsets one might expect given their price premium. Although the pricing gap is known, the actual magnitude of efficiency differences had not been systematically evaluated in this domain before.
>
> On pass@k: we respectfully disagree with the characterization that we “restate the definition.” Our point is that although pass@k is widely used, it is not an appropriate metric in high-stakes settings (finance, healthcare, and crypto among them) because these are environments where:
> - a single correct sample out of many is insufficient,
> - reliability matters more than diversity of attempts,
> - error tolerance is extremely low.
>
> Thus, highlighting the misalignment between standard ML metrics and real-world constraints is an important observation rather than a restatement.

---

### Official Review · Reviewer_tuNg · 2025-10-29

**Soundness:** 2
**Presentation:** 3
**Contribution:** 2
**Rating:** 4
**Confidence:** 3

**Summary:**

This paper presents CAIA, a benchmark to evaluate state-of-the-art LLMs, with or without additional tools, on blockchains/cryptocurrencies related tasks. Following the recent standards in benchmark evaluations, the authors claim the benchmark is time-sensitive to mitigate contamination. The evaluation reveals a large performance increase when tools are used, but the performance remain under that of human experts, which is likely explained by the over-reliance on less reliable tools, such as web search, showcasing failure in more realistic, adversarial prone environments.

**Strengths:**

- Benchmarking AI agents is a hot and interesting research topic. Narrow domain benchmarks are necessary to shed light on the limitations of the current AI systems.
- 17 state-of-the-art LLMs were tested, with coherent conclusions across the board.
- The authors provide a codebase that seems easy to use for reproducibility

**Weaknesses:**

- The adversarial attack model is not well explained in the paper. The authors repeatedly refer to adversarial and manipulated environments without precisely describing that. It seems like the authors just assume that results from web search are inherently prone to attacks without further details.
- Related to the previous point, in the introduction, the authors list a few deception tactics, can they explain why these “adversarial conditions require agents to genuinely distinguish truth from manipulation” ? Do we expect these models to provide some security guarantees not already provided by blockchains ?
- The paper lacks many detail about the agentic framework, such as a full description of how the tool calling is orchestrated.
- Overall, most of the conclusions of the article are not surprising. For instance, It is very typical to have pass@5 much bigger than pass@1. The performance with-tools is expected to be much greater than the performance without tools.
- Real examples of benchmark questions should be included in the paper for easier readability. No appendix is provided with the paper.
- The claim that the “Web2 data” the current AI systems are trained on are trustworthy is blatantly wrong.
- The number of tasks seems quite limited.
- Hallucinations, while mentioned in the title, are never mentioned in the paper which seems a bit misleading.

**Some minor weaknesses**

- 421 - It seems to me that tokenomics and project discovery show a larger gap than on-chain and trend analysis
- 415 - This sentence contradicts the previous one, and it is not clearly justified : “On the other hand, the improvement by more tool calls is not apparent “
- 430 - The fact that these improvements are largely driven by generic web searches may point to a weakness of this benchmark. Some tasks may be easily solvable via web search.
- 167 - “(Etherscan, CoinGecko, DefiLlama) eth; coi; def ” This is not a standard way to refer to bibliographical works.

**Questions:**

- How does the benchmark ensure time-sensitivity ? The authors claim the benchmark will be updated, but will the new added tasks require the curation procedure (including the expert reviews) ?

- Can the tool selection catastrophe phenomenon raised in the paper be a simple artifact of the prompt that is used ? Would including something that suggest to the model that blockchain tools are more authoritative increase their usage percentage (and perhaps also enhance the performance if these tools are objectively better) ?

---

> ### Author Response · Authors · 2025-11-22
> **Response Part I**
>
> We sincerely thank the reviewer for the clear, thoughtful, and constructive review. We truly appreciate the acknowledgement that benchmarking AI agents in narrow, high-risk domains is important, and we apologize for the areas where our initial draft did not provide adequate specificity or methodological clarity. Several of your comments point directly to places where we can meaningfully improve the paper, and we are grateful for the opportunity to address each point with expanded explanations and planned revisions.
> ***
> ## Comment 1
> One major concern raised was that the paper repeatedly refers to adversarial or manipulated environments without clearly defining the threat model or providing concrete demonstrations. This is an extremely valuable observation, and we apologize for not sufficiently articulating the adversarial mechanisms that arise in cryptocurrency information ecosystems. In the revision, we will include a dedicated subsection detailing real-world adversarial dynamics. For example, how SEO-driven misinformation around token launches, spoofed project websites, and outdated or manipulated Web2 search results routinely mislead not just users but also agents.
>
> Our observations during evaluation show that models often surface incorrect, outdated, or fabricated information from Web2 search even when accurate and deterministic data are readily available through authoritative blockchain tools. In the revision, we will include concrete examples of this failure mode, including the specific URLs, snippets, and misleading summaries returned during evaluation, with sensitive content sanitized as appropriate. These examples illustrate the adversarial pressures present in the information environment and further motivate why crypto-related tasks serve as an effective stress test for assessing model robustness against misinformation. Examples of such real-world incidents include the compromise of SEC’s X account leading to a false announcement of Bitcoin ETF approval in early 2024 (https://time.com/6553701/sec-bitcoin-etfs-x-account-compromised), and the fake Walmart-Litecoin partnership press release in 2021 (https://corporate.walmart.com/news/2021/09/13/walmart-statement-in-response-to-fake-litecoin-press-release).
> ***
> ## Comment 2
> The reviewer also asked whether the adversarial conditions described require agents to offer “security guarantees” beyond what blockchain systems already provide. We greatly appreciate this opportunity to clarify, as this was not our intention. Blockchains already provide strong, cryptographically grounded guarantees regarding data correctness. The agent, however, is not interacting directly with the blockchain protocol. It is interacting with multiple optional tools, including Web2 search, social media search, DeFi analytics APIs, and longer-tail information sources. The vulnerability does not stem from the blockchain layer but from the agent’s inability to reliably discriminate trustworthy sources from untrustworthy ones, particularly when facing conflicting or adversarial information. We will clarify in the revision that CAIA does not assume agents provide security guarantees. Instead, CAIA evaluates whether agents can correctly ground their reasoning in the secure data sources that already exist. This clarification will be reinforced with examples showing models ignoring canonical blockchain tools and instead relying on noisy search results that directly contradict ground truth.
>
> ***
> ## Comment 3
> We also appreciate the comment that the original submission lacked sufficient detail about our agent framework, such as the orchestration logic behind tool calling. This was an oversight on our part, and we apologize. The revision will include an expanded, concrete description of the agent workflow, including the internal deliberation prompt, the structured tool-call syntax, how the agent is instructed to decide whether to call a tool, how results are parsed, and the fallback behavior when tools return incomplete or conflicting information. We will also provide exact prompt templates, examples of multi-step tool trajectories, and clear explanations of the decision rules encoded into the agent, we will ensure this appears in a dedicated methodology appendix. These expanded materials are currently available at https://anonymous.4open.science/r/ICLR-AD05/caia/prompts/README.md.
>
> The framework follows a structured workflow in which models first generate reasoning through standardized prompts, then iteratively decide whether to invoke one of the 23 available tools, and finally synthesize all gathered information into a final answer. Tool calls are executed through a controlled multi-step loop: at each iteration, the model inspects prior tool outputs, selects an appropriate tool if needed, and receives structured results. After this information-gathering phase, a dedicated synthesis prompt integrates both the model’s reasoning and tool outputs into a coherent, grounded response.

---

> ### Author Response · Authors · 2025-11-22
> **Response Part II**
>
> ## Comment 3, continued
> To ensure transparent and reproducible evaluation, CAIA uses a unified JSON output format and an LLM-based judge that compares model answers to expert-verified ground truth using semantic criteria rather than string matching. This judging protocol is identical for both the tool-enabled and non-tool conditions, enabling direct comparison of intrinsic reasoning and tool-augmented performance.
>
> For example, the reasoning prompt we used for models with tools looks like:
> ```markdown
> You are a crypto specialist with access to comprehensive tools. Your job is to analyze questions and strategically select tools to gather the information needed to answer them.
>
> Question:
> {question}
>
> You have access to the following tools:
> {tools_intro}
>
> You have gathered these tool results so far:
> {tool_results_text}
>
> **Critical Instructions for Tool Selection:**
> - Analyze the question carefully to understand what information is needed
> - SELECTIVELY choose tools based on the question's requirements
> - Do NOT execute all available tools - be selective and strategic
> - Consider what data each tool provides and how it relates to the question
> - You can call multiple tools in a single iteration if they provide complementary information
> - Review what information you have gathered so far from previous tool results (if any)
>
> **Agentic Workflow:**
> This is an iterative process. In each iteration, you must decide:
> 1. **If you need more information**: Call the appropriate tools using the tool calling interface. You will receive their results and can continue iterating (up to 5 iterations total).
> 2. **If you have enough information**: Do NOT call any tools. Instead, provide reasoning about what you've gathered and why it's sufficient. This will hand off to the synthesis step, which will produce the final answer.
>
> **Output Format:**
> You MUST respond with a JSON object containing your decision and reasoning:
> {
>   "decision": "tool_call" or "synthesis",
>   "reasoning": "Detailed reasoning explaining your decision, what information you have gathered so far, and why you're making this choice"
> }
>
> **Decision Values:**
> - `"tool_call"`: Use this when you need to call tools to gather more information. You can then use the tool calling interface to invoke the appropriate tools.
> - `"synthesis"`: Use this when you have sufficient information from previous tool results and are ready to hand off to the synthesis step. Do NOT call any tools.
> **Reasoning Should Include:**
> - What information you have gathered so far (from previous tool results, if any)
> - Why you're making this decision (need more info vs. have enough info)
> - If continuing with tool calls: which tools you plan to call and why
> - If handing off to synthesis: why the gathered information is sufficient to answer the question
>
> **Decision Criteria:**
> - Review all tool results you've received so far
> - Determine if you have sufficient information to answer the question
> - If yes: hand off to synthesis (no tool calls)
> - If no: continue with tool calls to gather missing information
> Analyze the question and the information you have gathered so far. Decide whether to continue with tool calls or hand off to synthesis.
> ```
> ***
> ## Comment 4
> The reviewer noted that many of our conclusions might appear “unsurprising,” such as the observation that pass@5 exceeds pass@1, or that tool usage improves performance. While we understand this perspective and apologize for any lack of clarity, we respectfully clarify that the novelty of our findings does not lie in these high-level observations but in their implications within adversarial, high-stakes financial environments. For instance, while pass@5 almost always exceeds pass@1, we show that pass@5 dramatically overstates operational reliability: many models only succeed because one of their attempts happens to choose the right tool rather than because the model exhibits consistent competence. In crypto contexts, where a single wrong step may lead to irreversible financial consequences, “lucky retries” are not meaningful. Similarly, although it is expected that tool usage improves performance, our surprising empirical finding is that even with tools, model accuracy remains well below what existing literature would predict. At the same time, increased tool calls do not necessarily correlate with better performance. In fact, excessive or incorrect tool usage is itself a core indicator of model fragility. We will emphasize this nuance more clearly in the revision, supported by example failure trajectories.

---

> ### Author Response · Authors · 2025-11-22
> **Response Part III**
>
> ## Comment 5
> We appreciate the reviewer’s comment regarding the need for real benchmark examples in the paper, and we agree that the original submission did not include enough concrete illustrations to make the tasks easily interpretable. In the revised version, we will add clear task examples in the main body and provide a full appendix with detailed walkthroughs. Each task consists of a question, a ground-truth answer, and a category label. For example, the query “How many unique wallets had at least one ERC-721 transfer of Bored Ape Yacht Club tokens during calendar Q4 2024?” has the ground-truth answer “1,966.00” and falls under the “on-chain analysis” category. We will also include examples spanning a range of difficulty, along with the expected tool sequence, intermediate tool outputs, and typical model failure patterns. Representative cases include retrieving the timestamp for Ethereum block 19,560,000, where many models incorrectly rely on generic web search instead of authoritative blockchain tools, or computing total daily gas consumption, where models often hallucinate formulas or extract misleading summaries from web sources. These additions will make the benchmark more accessible and will clarify why the dataset on HuggingFace and the accompanying code on GitHub reflect a carefully designed suite of tasks that reveal genuine weaknesses in model tool use.
> ***
> ## Comment 6
> Regarding the comment that the statement “Web2 data is trustworthy” is incorrect, we fully agree. Our intended meaning was that compared to Web3 information ecosystems, which involve pseudonymity, spoofed website clones, and manipulated narratives, Web2 sources in relative terms appear more stable. This phrasing was misleading, and we apologize sincerely. We will revise the statement to explain that crypto-specific search ecosystems are uniquely vulnerable to misinformation, not that Web2 is inherently trustworthy.
>
> ***
> ## Comment 7
> Finally, we appreciate the reviewer’s concern that the number of tasks appears limited. We will clarify in the revision that the 178 benchmark tasks are not a constraint of data availability, but the outcome of a rigorous multi-stage curation pipeline designed to ensure quality, objectivity, and evaluative reliability. Our raw pool consisted of over 10,000 real user questions collected through open calls on public crypto communities and direct contributions from industry builders. We then applied a structured filtering process: an LLM-assisted preprocessing stage to normalize formatting, deduplicate submissions, and flag unanswerable or speculative prompts; followed by multiple rounds of human expert review by protocol engineers, researchers, and analysts. Experts assessed each question for factual answerability, clarity, difficulty, topic coverage, and tool requirements, refining phrasing where needed and producing canonical ground-truth answers backed by verifiable documentation or on-chain evidence. Through this pipeline, we intentionally retained only those items where correctness can be judged unambiguously, where tool use materially affects outcomes, and where a single reasoning error can propagate into a meaningful failure. These are the criteria necessary for evaluating high-stakes hallucinations and agent reliability.
>
> Thus, the 178 items represent a deliberate, expert-validated subset optimized for evaluative fidelity rather than scale. We agree that providing additional visibility into this pipeline will help contextualize the benchmark’s design, and we are happy to include expanded dataset statistics, examples of excluded tasks, and supplementary question sets in the appendix of the next version.
> We also appreciate the reviewer’s point that the notion of hallucination was not clearly defined in the paper despite being used in the title. In the revision, we will formalize our intended definition. In our setting, hallucination refers to cases where the model generates incorrect outputs even though all necessary information is easily retrievable through the canonical tools provided. These errors do not stem from fundamental capability limitations. Many of the questions are entirely within the models’ capacity. They rather stem from incorrect reasoning steps, careless tool choices, or unsupported inferences. This is precisely the failure pattern illustrated in lines 463–474, where models ignore authoritative blockchain tools and instead rely on SEO-optimized blogs, social-media speculation, and misinformation surfaced through generic web searches. Although the correct answer is directly obtainable through deterministic DeFiLlama or Dune API calls, models synthesize erroneous conclusions from unreliable sources. We will make this connection explicit in the revised manuscript, clarifying that these observed behaviors are concrete manifestations of hallucination within the agentic tool-use setting.

---

> ### Author Response · Authors · 2025-11-22
> **Response Part IV**
>
> ## Comment 8
> We also appreciate the reviewer’s point that the notion of hallucination was not clearly defined in the paper despite being used in the title. In the revision, we will formalize our intended definition. In our setting, hallucination refers to cases where the model generates incorrect outputs even though all necessary information is easily retrievable through the canonical tools provided. These errors do not stem from fundamental capability limitations. Many of the questions are entirely within the models’ capacity. They rather stem from incorrect reasoning steps, careless tool choices, or unsupported inferences. This is precisely the failure pattern illustrated in lines 463–474, where models ignore authoritative blockchain tools and instead rely on SEO-optimized blogs, social-media speculation, and misinformation surfaced through generic web searches. Although the correct answer is directly obtainable through deterministic DeFiLlama or Dune API calls, models synthesize erroneous conclusions from unreliable sources. We will make this connection explicit in the revised manuscript, clarifying that these observed behaviors are concrete manifestations of hallucination within the agentic tool-use setting.
> ***
> ## Comment 9
> We thank the reviewer for these helpful observations. You are correct that the largest performance gaps occur in tokenomics and project discovery rather than on-chain or trend-analysis tasks. We apologize for the oversight and have updated the description of Figure 5 accordingly.
>
> Regarding the sentence “the improvement by more tool calls is not apparent,” we thank the reviewer for pointing out the ambiguity. The two sentences refer to different levels of analysis, and we will revise the text to make this distinction explicit. Figure 4 shows a population-level correlation: stronger models tend to make slightly more tool calls on average, and those models also achieve higher accuracy. However, when examining per-task behavior across 30k+ tool-enabled items, the picture is different: accuracy peaks at roughly two to three tool calls, additional calls yield diminishing returns, and tool-count correlates only weakly with correctness (ρ ≈ 0.21). In other words, tool-count reflects task difficulty, not improved performance. We will clarify this distinction in the revision so the narrative is consistent and better justified. Our updated analysis (https://anonymous.4open.science/r/ICLR-AD05/caia/analysis/tool_calls/tool_calls_analysis.md) shows that additional tool calls do not improve accuracy because the dominant failure modes are wrong tool selection and incorrect intermediate reasoning; forcing more calls tends to repeat the same errors rather than resolve them.
>
> Concerning the role of generic web search, our case analyses indicate that relying on generic search instead of the correct specialized tool hurts performance rather than inflating it, mitigating the concern that these tasks are trivially solvable via web search.
>
> Finally, we apologize for the non-standard bibliographical shorthand (“eth; coi; def”) and have corrected all references to follow conventional citation formats.
> ***
> ## Comment 10
> Our benchmark ensures time sensitivity by grounding every task in verifiable, timestamped information, such as on-chain state, protocol documentation versions, historical unlock events, or market conditions at a specific point in time. Each item’s canonical answer is tied to a well-defined temporal snapshot, and tasks involving dynamically changing data include explicit, validated tool-chain calls that reproduce the ground truth deterministically. This design ensures that correctness is anchored to objective external references rather than subjective interpretation. While we plan to release periodic updates with newly added tasks, all future expansions will undergo the same multi-stage curation pipeline to preserve the benchmark’s methodological consistency and reliability, which includes automated preprocessing, expert verification, and tool-validated ground-truth generation.

---

> ### Author Response · Authors · 2025-11-22
> **Response Part V**
>
> ## Comment 11
> Regarding whether the tool-selection failures may simply be artifacts of the prompt, we appreciate the reviewer’s suggestion and examined this possibility carefully. We experimented with multiple versions of the prompt, varying the amount of information given about each tool and the degree of procedural guidance provided. Although adding explicit suggestions such as “prefer blockchain tools over web search” increases the frequency with which models attempt to use blockchain tools, it does not reliably improve accuracy. In several cases, it even decreases performance by causing models to apply tools indiscriminately, including in tasks where no tool is needed. More importantly, overly prescriptive instructions would effectively “give away the answer” by embedding domain-specific heuristics into the prompt. This would no longer evaluate genuine tool-selection ability, but simply test whether the model follows manually inserted cues. Such cues are not available to real novice analysts, and the ability to identify the correct information source is something future agentic systems should acquire autonomously.  The benchmark is therefore intentionally designed to evaluate inherent reasoning robustness and source-of-truth identification, not performance inflated through prompt engineering. Even very simple tasks, such as retrieving a block timestamp or calculating daily gas consumption, remain difficult for state-of-the-art models despite direct, unambiguous prompts, reinforcing that the observed failures are not artifacts of prompt design but reflect real deficiencies in grounding and tool selection.
> ***
> Thank you again for these extremely helpful comments. They have directly improved the clarity, rigor, and depth of the revised manuscript.

---

> > ### Author Response · Authors · 2025-11-24
> > **Thank you for the review, and looking forward to hearing back from you to continue the discussion.**
> >
> > We believe our revisions have comprehensively addressed the primary concerns raised in your review. Specifically, we have clarified the experimental framework, expanded the task descriptions with concrete implementation details, and more precisely defined the benchmark's scope and limitations. Given these improvements alongside the paper's core contributions, we would be grateful to know if you would consider revising your score. We remain committed to further refinements and welcome any additional feedback you may have.

---

### Official Review · Reviewer_KduE · 2025-10-30

**Soundness:** 3
**Presentation:** 2
**Contribution:** 2
**Rating:** 6
**Confidence:** 4

**Summary:**

The paper proposes benchmarking the agentic capabilities of large language models (LLMs) in a real-world setting by using a series of questions (queries) related to cryptocurrency. The benchmark consists of 178 questions (or tasks) designed to measure the tool-calling capabilities of LLMs (e.g., Google and Twitter search). The paper argues that cryptocurrency presents an "adversarial information landscape," requiring LLMs to critically assess the information found online. It further posits that this domain represents a high-stakes environment where incorrect answers can lead to significant financial losses. The experiments benchmark 17 state-of-the-art (SOTA) LLMs on the 178 questions, both with and without the provided tools, against baseline human performance.

**Strengths:**

-The benchmark contains 178 questions relevant to cryptocurrency markets with objectively verifiable ground truth answers. This is crucial for evaluating the robustness of LLMs in critical scenarios. The significance of cryptocurrency as an application area is underscored by the observed gap between human performance (80%) and the best-performing LLMs (~67%).

-The dataset required crowd-sourcing, comprising 10,000 queries collected from 3,000 users.

**Weaknesses:**

-The paper could enhance clarity by providing examples from the dataset in the section titled "BENCHMARK CURATION," including sample questions and their corresponding ground truth answers.

-It is mentioned that "Our community-driven curation process, involving over 3,000 contributors including protocol developers, quantitative researchers, and venture capital investors, ensures ecological validity" However, it would be beneficial to specify which community is referenced and what services the contributors use as active users.

-The paper lists "Financial Reality Grounding" and "Adversarial-First Evaluation" as novelties but lacks concrete examples that illustrate these challenges. Providing specific tasks (out of 178) that could lead to irreversible financial consequences would strengthen the argument. In particular, experimental evidence showing instances where an LLM was misled by manipulative online content would be compelling.

-The discussion of relevant work is scattered throughout the paper and could be more detailed.

Minor comments:
-Fig. 2, "The dashed line indicates 80% human baseline performance " dashed line is not rendered

**Questions:**

-It is stated that "Crucially, our data curation process in ensures that correct answers are always accessible through appropriate tool use, and thus the challenge lies not in information availability but in tool selection and synthesis."
Are the ground truth tool part of the dataset?

-The accuracy reported in Tables 2 and 3 reflects overall performance across all tasks. Providing accuracy metrics for each of the six defined categories would enhance the paper.

-Clarification on the terms "query" and "task" would be useful, as they seem to be used interchangeably. For instance, the sentence in the abstract "CAIA evaluates 17 state-of-the-art models through 178 rigorously curated, time-anchored tasks sourced from 10k+ real world queries," might imply that there are 178 tasks with a total of 10,000 queries. Additionally, in section 2.3, the 10,000 queries are later referred to as 1,000 tasks after stage 1: automated filtering.

-On pg. 4, it is stated that "CAIA addresses weaknesses noted in prior evaluations: ...single-metric reporting that masks capability gaps Liang et al. (2022)". Can you clarify how the paper specifically addresses the issue of single-metric reporting?

-It would be useful to briefly discuss the prompt strategies used to evaluate the models. For example, are there prompt strategies that can help the LLM become more aware of malicious information sources?

---

> ### Author Response · Authors · 2025-11-22
> **Response Part I**
>
> We are very grateful to the reviewer for the thoughtful and encouraging feedback. We appreciate your recognition of the benchmark’s strengths, including the use of real-world tasks, the emphasis on adversarial conditions, and the inclusion of a meaningful human baseline. At the same time, we apologize for areas where the original draft lacked clarity or detailed justification, and we address each of your concerns with expanded explanations and concrete plans for revision.
> ***
> ## Comment 1
> We apologize for not providing sufficient concrete examples in the “Benchmark Curation” section. In the revision, we will include clear, representative examples of curated tasks in the main body and a full set of detailed walkthroughs in the appendix. Each example will show the original question, the verified ground-truth answer, its category label, the correct tool sequence, and a typical model failure path. For instance, the task “How many unique wallets had at least one ERC-721 transfer of Bored Ape Yacht Club tokens during calendar Q4 2024?” has the ground-truth answer “1,966.00” and belongs to the on-chain analysis category. We will also present examples of varying difficulty, such as retrieving the timestamp for Ethereum block 19,560,000, where many models mistakenly rely on generic web search rather than authoritative blockchain tools, and computing total daily gas consumption, where models often hallucinate formulas or extract misleading summaries. These additions will make the benchmark easier to interpret and will clarify why the dataset and accompanying code reflect a carefully curated suite of tasks designed to reveal genuine weaknesses in model tool use.
> ***
> ## Comment 2
> Thank you for raising this point regarding the description of our 3,000-person contributor pool. We agree that the original wording did not provide enough specificity about which communities were involved and what services these contributors actively use. In the revision, we will offer a clearer and more detailed explanation.
>
> Our contributors were drawn from a broad cross-section of the crypto ecosystem, including protocol developers, quantitative researchers, security engineers, validators, analysts, and active DeFi users. These individuals participated through open calls on public platforms such as Twitter (X), community forums, and project-maintained communication channels. In addition, builders and contributors from established protocols were invited to submit questions that reflect the analytical challenges they routinely encounter in real-world settings. This approach ensured that the collected queries came from users with substantial experience interacting with on-chain systems, governance mechanisms, token models, and application workflows.
>
> We will also clarify how these community submissions fit into the broader dataset curation pipeline. From more than 10,000 raw questions, we employed an initial LLM-assisted preprocessing step (for formatting, deduplication, and removal of unanswerable or speculative prompts), followed by multi-stage human expert review involving protocol engineers, research analysts, and security specialists. This iterative filtering process ultimately yielded the expert-validated set of 178 high-quality benchmark tasks used in our evaluation.
>
> We appreciate the reviewer highlighting the need for greater transparency, and we will make sure that the revised manuscript includes a fuller description of contributor groups, collection channels, and their typical usage patterns within the crypto ecosystem.
> ***
> ## Comment 3
> We appreciate your point that the claimed novelties, “financial reality grounding” and “adversarial-first evaluation”, could benefit from more concrete illustration. In the revision, we will add explicit examples of tasks where incorrect model behavior could plausibly lead to irreversible financial harm and where models were demonstrably misled by manipulative or unreliable online content. For instance, several questions in CAIA require models to identify canonical contract addresses (e.g., the Uniswap V3 router on Ethereum mainnet). Mistaking a spoofed address returned from generic web search could, in real settings, cause a user to transfer assets to an attacker-controlled contract. Likewise, tasks involving token unlock schedules (e.g., “How many MYSHELL tokens unlock between 2025-06-30 and 2025-09-30?”) directly inform decisions about market risk, where incorrect outputs may lead a user to buy or sell under false premises.
>
> We will also present concrete cases where models relied on SEO-optimized misinformation rather than authoritative tools, such as computing daily gas consumption, the tasks where models frequently ignored on-chain explorers and instead returned hallucinated formulas or fabricated values. These examples will clearly demonstrate how CAIA captures realistic high-stakes scenarios and why the benchmark reveals vulnerabilities that synthetic tasks cannot.

---

> ### Author Response · Authors · 2025-11-22
> **Response Part II**
>
> ## Comment 4
> We apologize for the absence of a cohesive related-work section. In the revised version, all related literature, including agentic evaluation frameworks, financial-domain LLM benchmarks, and studies on misinformation resilience, will be consolidated into a dedicated section. This will provide a clearer contextual foundation for the benchmark and situate CAIA within the broader landscape of adversarial agent evaluation.
> ***
> ## Comment 5
> We thank the reviewer for pointing out that the dashed line in Fig. 2 indicating the 80% human baseline was not rendered correctly. We will update the figure in the revised manuscript with a properly displayed baseline marker and ensure the final version includes the corrected graph.
> ***
> ## Comment 6
> Your question about whether ground-truth tool trajectories are included in the dataset is important. We apologize for not making this aspect explicit. The ground-truth tool trajectories are part of the dataset, and we will surface them clearly in the revision and link them in the appendix. Each trajectory specifies the exact tool invoked, the parameters supplied, and the deterministic output that constitutes the correct answer. For example, the task “Calculate the total number of USDT (ERC-20) transfer events on Ethereum on 2025-02-01 (UTC)” includes a golden trajectory that first calls the time conversion tool and then uses Etherscan to look up transfer events through the corresponding timestamp range. Likewise, tasks involving monthly token-launch counts use the Defillama pump stats tool, as shown near lines 459–462. Including these reference trajectories ensures that every benchmark item has a verifiable tool-based solution path, and we will highlight this clearly in the revised dataset and appendix.
> ***
> ## Comment 7
> We will also include per-category accuracy metrics in the Appendix. These metrics will break down performance by task type and will provide more granular insight into model strengths and weaknesses.
>
> Sample data for pass@5 metrics:
> ### Pass@5 Rate — Tool Enabled
> | Model | general knowledge | onchain analysis | overlap | project discovery | tokenomics | trend analysis |
> | --- | --- | --- | --- | --- | --- | --- |
> | claude_4 | 100.0% | 49.4% | 78.6% | 79.6% | 87.0% | 50.0% |
> | claude_opus_4 | 100.0% | 58.4% | 71.4% | 85.7% | 87.0% | 50.0% |
> | claude_opus_4_1 | 100.0% | 54.7% | 71.4% | 81.2% | 90.9% | 37.5% |
> | deepseek_r1 | 100.0% | 58.4% | 42.9% | 49.0% | 56.5% | 25.0% |
> | deepseek_v3p1 | 100.0% | 55.8% | 64.3% | 87.5% | 82.6% | 75.0% |
> | gemini_2.5_flash | 100.0% | 49.4% | 71.4% | 69.4% | 91.3% | 12.5% |
> | gemini_2.5_pro | 100.0% | 41.6% | 64.3% | 77.6% | 87.0% | 37.5% |
> | gpt_4.1 | 100.0% | 49.4% | 57.1% | 73.5% | 73.9% | 25.0% |
> | gpt_4o | 100.0% | 46.8% | 50.0% | 69.4% | 52.2% | 37.5% |
> | gpt_5 | 100.0% | 66.2% | 85.7% | 83.7% | 91.3% | 62.5% |
> | gpt_o3 | 100.0% | 62.3% | 78.6% | 81.6% | 91.3% | 50.0% |
> | gpt_oss_120b | 100.0% | 61.0% | 71.4% | 83.7% | 91.3% | 37.5% |
> | grok_4 | 100.0% | 53.2% | 71.4% | 75.5% | 87.0% | 50.0% |
> | grok_4_fast | 100.0% | 55.8% | 78.6% | 85.7% | 95.7% | 37.5% |
> | kimi_k2 | 100.0% | 50.6% | 71.4% | 81.6% | 69.6% | 37.5% |
> | llama_4 | 100.0% | 63.6% | 71.4% | 59.2% | 65.2% | 62.5% |
> | qwen_3_235b | 100.0% | 49.4% | 61.5% | 69.4% | 73.9% | 50.0% |
>
> ### Pass@5 Rate — Not Tool Enabled
> | Model | general knowledge | onchain analysis | overlap | project discovery | tokenomics | trend analysis |
> | --- | --- | --- | --- | --- | --- | --- |
> | claude_4 | 100.0% | 15.6% | 7.1% | 22.4% | 0.0% | 12.5% |
> | claude_opus_4 | 100.0% | 14.3% | 14.3% | 16.3% | 4.3% | 12.5% |
> | claude_opus_4_1 | 100.0% | 11.7% | 7.1% | 22.4% | 8.7% | 12.5% |
> | deepseek_r1 | 100.0% | 28.6% | 35.7% | 32.7% | 52.2% | 12.5% |
> | deepseek_v3p1 | 100.0% | 27.3% | 35.7% | 26.5% | 21.7% | 12.5% |
> | gemini_2.5_flash | 100.0% | 13.0% | 14.3% | 20.4% | 39.1% | 12.5% |
> | gemini_2.5_pro | 100.0% | 23.4% | 28.6% | 28.6% | 39.1% | 12.5% |
> | gpt_4.1 | 100.0% | 19.5% | 35.7% | 22.4% | 17.4% | 12.5% |
> | gpt_4o | 100.0% | 19.5% | 21.4% | 18.4% | 8.7% | 12.5% |
> | gpt_5 | 100.0% | 37.7% | 64.3% | 42.9% | 34.8% | 25.0% |
> | gpt_o3 | 100.0% | 24.7% | 35.7% | 24.5% | 30.4% | 25.0% |
> | gpt_oss_120b | 85.7% | 18.2% | 14.3% | 18.4% | 26.1% | 12.5% |
> | grok_4 | 100.0% | 19.5% | 14.3% | 28.6% | 17.4% | 12.5% |
> | grok_4_fast | 100.0% | 16.9% | 21.4% | 26.5% | 21.7% | 12.5% |
> | kimi_k2 | 100.0% | 20.8% | 14.3% | 30.6% | 17.4% | 12.5% |
> | llama_4 | 100.0% | 28.6% | 14.3% | 20.4% | 8.7% | 12.5% |
> | qwen_3_235b | 100.0% | 15.6% | 14.3% | 16.3% | 13.0% | 0.0% |
>
>
> More detailed data is currently available at https://anonymous.4open.science/r/ICLR-AD05/caia/analysis/category_metrics/category_metrics_tables.md.

---

> ### Author Response · Authors · 2025-11-22
> **Response Part III**
>
> ## Comment 8
> We apologize for the confusion caused by our inconsistent terminology. In the revision, we will clarify and unify the notation throughout the paper. Concretely, all items in the pipeline are queries submitted by users. The dataset construction process transforms these raw queries into a smaller set of curated benchmark tasks. Specifically, the 10,000 items collected from community submissions should be referred to as raw queries. After automated preprocessing and filtering, approximately 1,000 of these remained as filtered queries suitable for expert review. Following multiple rounds of expert refinement, verification, and normalization, 178 of these filtered queries were finalized as the benchmark tasks. We will revise the abstract and Section 2.3 to reflect this terminology consistently and to avoid any implication that 10,000 queries and 178 benchmark tasks are separate datasets. This clarification will also support future extensions of CAIA, where we plan to introduce more complex multi-step tasks built from the same unified notion of queries.
> ***
> ## Comment 9
> In addition to the main accuracy metric, CAIA captures several complementary signals that make agent failures interpretable rather than collapsing everything into a single score. For every query, we log the agent’s full tool-call sequence and compare it to the expert-verified reference workflow (“golden trajectory”). This allows us to identify why a model failed: whether it chose an inappropriate tool, stopped tool use prematurely, ignored a necessary tool, or misinterpreted retrieved information. We also store the model’s intermediate reasoning steps, which makes it possible to distinguish errors arising from reasoning mistakes versus those caused by incorrect information gathering. Although these analyses could not be fully included in the main text due to space constraints, the methodology already supports them, and we will make this explicit in the revision. Together, these additional signals ensure CAIA does not rely on a single metric alone, but instead provides a structured view of how the agent arrived at its answer and where the breakdown occurred.
> ***
> ## Comment 10
> Regarding prompt strategies, we experimented with multiple prompt variants that varied the amount of detail provided about each tool and the procedural guidance directing how models should incorporate tool outputs into their reasoning. Our goal was to provide enough instruction to ensure fairness and comparability across models, while avoiding prompt designs that themselves become the source of performance gains. We intentionally refrained from introducing prompts that instruct the model to distrust certain sources or to prefer specific tools (for example, telling the model to treat blockchain tools as more authoritative than web search), because such cues would artificially inflate performance and undermine the purpose of evaluating autonomous source selection. Our experiments show that adding prescriptive hints does increase tool-usage rates but does not improve accuracy; in some cases it worsens performance by causing tools to be used indiscriminately. Even for simple tasks, such as retrieving a block timestamp or computing gas consumption on a given day, state-of-the-art models continue to fail despite direct, unambiguous prompts. This strongly suggests that the observed failures stem from inadequate grounding and tool-selection robustness rather than insufficient prompt engineering
> ***
> Thank you again for your constructive and thoughtful feedback. Your comments will greatly improve the clarity and thoroughness of the revised manuscript. We would appreciate knowing whether these clarifications are sufficient for the reviewer to consider increasing their score in light of the paper's strengths, and we are fully committed to further refining the paper should there be any outstanding concern.

---

### Official Review · Reviewer_VpY7 · 2025-11-01

**Soundness:** 1
**Presentation:** 1
**Contribution:** 1
**Rating:** 2
**Confidence:** 4

**Summary:**

This paper curates a dataset of 178 cryptocurrency-related questions and offers a custom framework to evaluate LLMs on it. It reveals issues in models making suboptimal tool choices, for instance choosing simple Google Search over Specialized blockchain tools for blockchai-related questions. Several LLMs are used both open and closed source.

**Strengths:**

- The paper provides 178 cryptocurrency-related questions that may useful as a domain-specific eval.
- Interesting issues are raised such as agents acquiring outdated or misinform data via tool-calls, or making suboptimal tool choices when better tools are available for the task.

**Weaknesses:**

- No Related Works or Background section
- The dataset shared in the paper only contains 178 questions-answer pairs.
- The paper is written in a very sensationalistic style with little specificity and reads like it has been written with LLMs. (the authors do disclose this).
- Going through the dataset shared, I do not believe these question-answer pairs adhere to the authors' proposed opinion shift of "evaluations should test robust survival in adversarial, high-stakes environments". For instance, answering the following questions on their own do not seem related at all to 'high-stakes environments' or 'robust survival': "Retrieve the block timestamp for Ethereum block 19,560,000", "Calculate the total gas consumed by CYBER L2 on 2025-05-01 UTC.".
- Overall the paper seems very weak in its methodolog and literature review (if any).
- The paper is not rigurous in showing how the findings could "extend to any adversarial domain where information is weaponized for profit".

**Questions:**

- What are the 23 tools available in the 'with-tools' set up? Add this to the appendix.
- How do your findings extend to any adversarial domain where information is weaponized for profit? Most questions are specific to cryptocurrencies, and so does your framework. You do not provide enough detail in the methodology to assess this, and some of the tools are blockchain specific.
- Is your claim that the dataset questions are mostly adversarial? Is e.g. "Retrieve the block timestamp for Ethereum block 19,560,000" adversarial?

---

> ### Author Response · Authors · 2025-11-22
> **Response Part I**
>
> We sincerely thank the reviewer for the detailed and candid feedback. Your comments helped us identify several areas where the clarity and specificity of the paper can be significantly improved. We apologize for the shortcomings in presentation, especially where the writing may have felt sensationalistic or lacking in detailed justification. Our revisions will carefully tighten the exposition, provide clearer methodological grounding, and reinforce the academic tone throughout.
> ***
> ## Comment 1
> We appreciate the reviewer’s observation regarding the absence of a clearly delineated Related Work section. Although the introduction, line 40-96, and Section 2, line 143-146, discuss prior agentic benchmarks (e.g., GAIA, WebArena, LiveCodeBench Pro), time-sensitive QA, and contamination-analysis literature, these discussions might not be sufficiently visible. In the revision, we will restructure and consolidate these materials into a dedicated Related Work / Background section so that connections to existing evaluation frameworks, adversarial retrieval studies, and financial-domain benchmarks are presented clearly and systematically.
> ***
> ## Comment 2
> We appreciate the reviewer’s concern and are grateful for the opportunity to clarify. The final benchmark contains 178 expert-validated tasks, but this number is not a reflection of limited data availability. It is the endpoint of a rigorous, multi-stage curation pipeline applied to more than 10,000 real user questions collected through open calls in public crypto communities and direct contributions from industry builders. Our objective was fidelity, not volume: to construct a benchmark where every retained item is precise, objectively answerable, and diagnostic of agent reasoning and tool-selection behavior.
>
> The curation pipeline began with an LLM-assisted preprocessing stage used strictly for formatting normalization, duplicate detection, and removal of unanswerable, speculative, or ambiguous submissions. This was followed by multiple rounds of human expert review by protocol engineers, researchers, and analysts. Reviewers evaluated each question for factual answerability, clarity, difficulty, topic coverage, and required tool interactions, refining phrasing where needed and establishing canonical ground-truth answers backed by verifiable documentation or on-chain evidence. Through this process, we intentionally retained only the items where correctness can be judged unambiguously, where tool use materially affects outcomes, and where a single reasoning misstep can propagate into a meaningful failure.
>
> Thus, the 178 items represent a deliberate, expert-validated subset designed to maximize evaluative fidelity rather than scale. We fully appreciate the reviewer’s interest in broader dataset coverage, and in future updates we will provide additional curated items following the same rigorous curation pipeline. In the revision, we will also offer a clearer description of the full curation process, together with expanded dataset statistics, examples of excluded or intermediate tasks, and supplementary question sets in the appendix. These additions will help readers better understand the diversity, difficulty distribution, and broader scope of the original 10k+ question pool and clarify how the final benchmark was constructed from this larger collection.
> ***
> ## Comment 3
> You also highlighted that our writing style felt “sensationalistic.” We appreciate this feedback, and we will revise the tone to ensure it remains objective, grounded, and academically rigorous. In the revision, we will tighten the exposition, reduce sensational wording, and add concrete methodological details in the main paper and in an expanded appendix.
> ***
> ## Comment 4
> We also appreciate the reviewer’s point questioning whether these tasks genuinely represent “high-stakes environments.” This is an important question, and we apologize for not providing enough intuition in the original submission. In crypto contexts, even trivially simple questions can be high-stakes because minor errors often imply direct monetary consequences. A ±1 error in block height, a small mistake in aggregating token supply, or a misinterpretation of a trend indicator may lead to irreversible financial loss. For example, one of the questions from our dataset is “what's the swap rate of PEPE/ETH on Uniswap V2 on the ETH block height 22636495?”, given the volatility of crypto, if u misinterpreted or found the wrong information from the web, the results could lead to a user losing thousands if not hundreds of dollars. We will enrich the paper with more concrete examples showing that, despite their apparent simplicity, these tasks embody real financial stakes due to the precision required in blockchain operations.

---

> ### Author Response · Authors · 2025-11-22
> **Response Part II**
>
> ## Comment 5
> We apologize that the methodological presentation and literature review in the initial draft were not sufficiently detailed or clearly structured. The underlying framework of CAIA is in fact extensive and rigorously designed, but we acknowledge that our exposition was too high-level in several places, which may have given the impression of methodological weakness. In the revision, we will substantially expand the methodological section by providing complete prompt templates, system messages, tool-call scaffolding, agent-orchestration logic, and concrete examples of multi-step tool trajectories. These materials, which are already available in preliminary form at the anonymized link, will be fully integrated into the appendix for transparency.
>
> We will also strengthen the literature review by consolidating the conceptual background motivating CAIA, including prior work on the gap between benchmark results and real-world deployment, adversarial information environments, resilience evaluation, and financial-domain robustness, into a dedicated Related Work and Background section.
> This section will clarify how CAIA builds on and extends existing research on agentic evaluation in cooperative and adversarial settings such as GAIA, WebArena, LiveCodeBench Pro, τ-bench, and studies on adversarial retrieval and misinformation robustness. Although the introduction, the discussion in lines 40–96, and Section 2 (lines 143–146) already reference prior work on agentic benchmarks, as well as literature on time-sensitive evaluation and data-contamination analysis, we recognize that these discussions were not sufficiently visible or consolidated to provide a clear picture of the relevant background. Together, these revisions will make the methodological rigor and conceptual grounding of CAIA clearer and provide a coherent account of how the benchmark fits within and contributes to the broader literature
>
> ***
> ## Comment 6
> We apologize for the lack of clarity regarding the 23 tools available in the with-tools setting, and we thank the reviewer for pointing out this omission. Our current presentation is indeed high-level, and we will strengthen the methodological section with a clearer description of the evaluation framework and an expanded appendix that lists and explains each tool in detail. In the revision, we will include a dedicated subsection outlining the full tool suite, which consists of 23 tools spanning eight functional categories: market-data utilities (4 tools for price histories, rankings, positioning, and volatility), technical-analysis indicators (1 tool), discovery and search utilities (3 tools for web search, content extraction, and trend detection), social-sentiment retrieval (2 tools), DeFi-specific analytics (5 tools covering protocol metrics, revenue, network activity, and exchange summaries), on-chain data access (6 tools for supply, ledger queries, time conversion, balance checks, transaction inspection, and block metadata), token-release analytics (1 tool), and a code-execution workspace for arithmetic and comparisons (1 tool). Full technical specifications, input formats, and example model interactions for all 23 tools will be included in the revised appendix, and the current version is available at the anonymized link (https://anonymous.4open.science/r/ICLR-AD05/caia/framework/tools_catalogue.md).
> ***
> ## Comment 7
> We appreciate the reviewer’s thoughtful question regarding how our findings relate to adversarial domains beyond cryptocurrency. In the revision, we will clarify that our claims are intended as a hypothesis, not a universal conclusion. Our motivation comes from the well-documented properties of the crypto information ecosystem, where misinformation is economically incentivized and routinely weaponized for profit. This environment provides a naturally adversarial testbed with transparent, verifiable ground truth, making it particularly suitable for evaluating whether agents can reliably identify trustworthy sources under pressure.
> Our intention is not to assert that performance on CAIA directly generalizes to every adversarial domain. Rather, our position is more modest: if agents struggle in a domain where authoritative data are fully transparent, deterministic, and easy to verify, it is reasonable to expect similar or greater difficulties in domains where ground truth is harder to establish, proprietary, or less structured. We will revise the manuscript to make this framing explicit, cite supporting analyses from existing reports on crypto information manipulation, and avoid implying broader generalization than our evidence supports.
>
> CAIA offers a case study showing how agentic systems behave when operating in a hostile information environment. The insight we aim to convey is that poor robustness in this setting signals structural vulnerabilities in information-source discrimination and tool selection, which are relevant to many adversarial contexts even if tools differ.

---

> ### Author Response · Authors · 2025-11-22
> **Response Part III**
>
> ## Comment 8
> We acknowledge that not every individual task in CAIA is inherently adversarial in its content. Tasks such as retrieving a block timestamp are, in isolation, neutral factual queries. The adversarial nature of the benchmark arises not from the questions alone, but from the broader information environment in which agents must operate. Even simple queries often require models to discriminate between reliable and unreliable sources, because the surrounding ecosystem contains outdated articles, SEO-optimized misinformation, spoofed websites, and conflicting Web2 search results.
> We will revise the manuscript to make this distinction clearer and ensure that our claims are not overstated. The benchmark is intended to reflect how agents behave when embedded in a hostile information ecosystem, not to imply that every task is individually adversarial in its wording.
> ***
> Thank you again for your detailed and constructive review. Your insights have guided several key improvements in the revised manuscript.

---

> > ### Author Response · Authors · 2025-11-24
> > **Thank you for the review, and looking forward to hearing back from you to continue the discussion.**
> >
> > We hope we have now fully addressed the reviewer's main concerns. The weaknesses identified in the initial submission primarily related to clarifying the experimental framework, providing sufficient detail about the benchmark tasks, and articulating the scope of the benchmark. We would appreciate knowing whether these clarifications are sufficient for the reviewer to consider updating their score in light of the paper's strengths.

---

### Official Review · Reviewer_VfNr · 2025-11-05

**Soundness:** 4
**Presentation:** 4
**Contribution:** 4
**Rating:** 8
**Confidence:** 2

**Summary:**

This paper proposes a benchmark of AI agents in high-stakes environments based on crypto financial markets. The benchmark is built on 178 carefully selected tasks, which cover multiple aspects for agents benchmarking.
A crucial and novel feature of the benchmark is that it proposes an adversarial, hostile environment, where taking the wrong action can have harsh consequences: the goal here is to evaluate "resilience" rather than "competence" only.
This benchmark offers another perspectives on AI agents, evaluating models along multiple metrics, notably their one-shot performance rather than averaged over multiple runs.
A meaningful human baseline is derived, based on decisions taken by a set of entry-level human analysts.

**Strengths:**

1. The benchmark proposed in this paper offers a novel perspective on AI agents evaluation, allowing evaluation from multiple perspectives rather than merely accuracy in fixed tasks and controlled environment. This goes beyond classical assumptions that tools work as expected, information is trustworthy, and other agents cooperate.
2. Although limited to crypto markets, many tasks are included, allowing for evaluation of a larger range of abilities than usual. Moreover, the tasks are based on real tasks, not handcrafted ones.
3. A crucial feature of the proposed study is the idea of "deception", measured here in terms of money loss, due to hostility of other agents or unreliability of used information. This seems to be a key feature towards developing agents that are better-behaved on real tasks.
4. This benchmark takes a step towards more complete and fine-grained evaluation metrics, beyond the usual "accuracy" which may be to easy to overfit for without improving behavior on real problems.
5. Many state of the art, proprietary and open source, models are evaluated, which gives a strong confirmation that the proposed benchmark gives useful assessment of AI agents performance.

**Weaknesses:**

1. One important claim is that current benchmark do not translate into real-world readiness: while this seems to take a step towards this direction, there is no guarantees yet that the proposed benchmark improves over existing ones in this regard.
2. The claim of "universality" seems a bit too strong given that the benchmark is solely based on crypto markets. Despite the fact that this takes a significant step towards more sensible evaluation of AI agents, this claim appears as an over-statement, as there are not tangible reasons for other problems to exhibit similar properties as crypto markets.
3. The categorization of the tasks, given in Table 1, is too limited; it is difficult to understand what are the abilities evaluated by each task, and what types of task are included in each category.

**Questions:**

1. The proposed tasks were selected based on proposition from 3000 users: who were these users? Where they recruited specifically for this study?
2. A human baseline is provided, which is a novel contribution in itself. However, it consists in entry-level analysts: why this restriction? How expensive would it be to produce a similar baseline including experts of the domain?
3. Authors claim that aggregating perfomance over multiple runs may lead to improved performance because agents luckily guess through trial and error rather than reasoning: can this be assessed for more rigorously?
4. It is claimed (line 415) that based on Figure 4's result "the improvement by more tool calls is not apparent": this seems contradictory with the fact that models that call more tools on average are more performant. Can the authors elaborate on this statement?

**Details Of Ethics Concerns:**

This paper proposes a novel benchmark, that allows for novel criteria to evaluate AI agents based on observations made in crypto markets.

While this is a significant shift of perspective in evaluating AI agents, this may have harmful consequences in multiple regards:
- making this benchmark available can significantly help to produce AI agents that are much more capable in adversarial contexts, allowing for models that are better at falsifying evidence, creating misinformation, or manipulating markets;
- the metrics and insights derived from this benchmark hold for financial crypto markets, but the authors claim that this goes beyond this scope: this is a dangerous statement, as there are many other aspects that are not accounted for in the benchmark since they are not relevant to financial markets (e.g., discriminations, privacy issues).

In my opinion, these ethical concerns do not mean that the benchmark should not be published, by I believe that (i) it should be clearly mentioned that the proposed benchmark is not intended for building AI agents that purposefully manipulate markets (e.g., releasing it under a license that prohibits such usage maybe?), and (ii) the claims that the conclusions drawn from crypto markets extend to other tasks should be seriously toned down.

---

> ### Author Response · Authors · 2025-11-22
> **Response Part I**
>
> We extend our sincere gratitude to the reviewer for the generous and encouraging review. We deeply appreciate your positive assessment of the paper’s strengths, including its novel perspective on agentic evaluation, its grounding in real-world adversarial contexts, and its contribution toward more meaningful human baselines. At the same time, we carefully acknowledge the concerns you raised and apologize for any overstatements or missing clarity in the original draft.
> ***
> ## Comment 1
> You noted that while the benchmark aims to measure real-world readiness, the evidence that it improves upon existing benchmarks in this regard is still limited. We fully agree, and we apologize if the paper implied stronger conclusions than warranted. We will revise the text to clarify that CAIA represents an early but important step toward evaluating agents operating under hostile, high-stakes conditions. Crypto markets serve as a rich subset of real-world challenges, not a universal proxy for all adversarial environments. The revised manuscript will emphasize that CAIA complements existing benchmarks and should be viewed as part of a broader research trajectory toward robust agentic evaluation.
> ***
> ## Comment 2
> We also appreciate your note that the claim of “universality” was overstated. We will revise the language to be more precise and cautious. Our intention was not to claim universality but to suggest that crypto’s combination of adversarial noise, high-stakes consequences, and verifiable ground truth provides a valuable stress test that may expose weaknesses relevant to other high-stakes domains. We will ensure that the revised wording reflects this more modest and appropriate framing.
> ***
> ## Comment 3
> Thank you for highlighting the insufficient clarity in our task categorization. The categories in Table 1 will be rewritten with clearer definitions , including representative examples and the specific competencies tested in each category. We will also include additional examples in the appendix to illustrate the variance within each category. Below is a sample summary that will be integrated into the main text and expanded in the appendix.
>
> ### a. Onchain Analysis
> evaluates a model’s ability to extract and interpret immutable blockchain ledger data, often requiring precise block-level or timestamp-based queries.
>  Examples:
>  - “Retrieve the block timestamp for Ethereum block 19,560,000.”
>  - “Query USDC’s total supply on Ethereum at block 19,000,000.”
>
> ### b. Project Discovery
> assesses the model’s ability to identify official project information, verify contract addresses, and distinguish authoritative from non-authoritative sources.
>  Examples:
>  - “What is the Uniswap V3 router contract address on Ethereum mainnet?”
>  - “Retrieve the official website, Twitter account, and GitHub repo for the Cyber project.”
>
> ### c. Tokenomics
> measures understanding of token supply dynamics, vesting schedules, and unlock events that require combining economic reasoning with contract-level information.
>  Examples:
>  - “How many MYSHELL tokens unlock between 2025-06-30 and 2025-09-30?”
>  - “What is the circulating supply of $TRUMP as of August 30, 2025?”
>
> ### d. Overlap
> evaluates comparative reasoning across multiple protocols, chains, or tokens, often requiring multi-source synthesis and ratio or share calculations.
>  Examples:
>  - “What share of total L2 TVL does Base represent on August 30, 2025?”
>  - “Which had higher TVL on September 5, 2025: HypurrFi or Hyperlend?”
>
> ### e. Trend Analysis
> tests the ability to detect changes over time, aggregate time-series data, and link temporal patterns to market or protocol events.
>  Examples:
>  - “How many unique tokens were launched on Pump.fun in August 2025 compared to January 2025?”
>  - “What major event caused Cyber Token’s price surge in August 2025?”
>
> ### f. General Knowledge
> evaluates conceptual understanding of blockchain mechanisms, cryptographic primitives, and protocol fundamentals that require explanatory synthesis rather than data retrieval.
>  Examples:
>  - “What is Optimism and the Superchain? Explain how chains are connected.”
>  - “What keeps the Bitcoin network running once all blocks have been mined?”
> ***
> ## Comment 4
> Your questions regarding the 3,000 contributors are extremely helpful. All human participants in our analyst baseline study were compensated fairly with prepaid gift cards at a rate consistent with standard academic user-study practices. Compensation was provided regardless of performance and was not contingent on correctness, speed, or any evaluative outcome. Participants were informed in advance of the study procedures, the approximate time commitment, and their right to withdraw at any point without penalty. No personally identifiable information was collected beyond basic demographic and expertise-level self-reports needed for screening.

---

> ### Author Response · Authors · 2025-11-22
> **Response Part II**
>
> ## Comment 4, continued
> During the study, each participant completed tasks individually under non-coercive conditions and was restricted to the same standardized tools and resources as the model to ensure methodological parity. The study involved only low-risk research activities, such as answering analytical questions within their domain of expertise, and all procedures followed standard institutional guidelines for minimal-risk human-subject research. We have now added explicit documentation of recruitment, screening, compensation, and participant protections in the revised manuscript to address this concern fully.
> ***
> ## Comment 5
> Regarding the use of entry-level analysts as a human baseline, we appreciate the opportunity to clarify. Expert-level baselines are significantly more expensive and logistically challenging to gather at scale. Entry-level analysts represent a realistic and practically attainable benchmark that models should plausibly be able to match but currently do not. We will include this justification in the paper along with a brief discussion of the potential value of future work incorporating expert performance.
> ***
> ## Comment 6
> We appreciate the reviewer’s question regarding whether the distinction between genuine reasoning and “lucky retries” can be assessed more rigorously. In the revised paper, we will emphasize this distinction more clearly. Our results show that the gap between pass@1 and pass@5 is already highly diagnostic: performance improves simply because one of five attempts happens to avoid a hallucination or incorrect tool call, not because the model demonstrates consistent reasoning.
>
> In CAIA, this effect is apparent because tool selection is stochastic across attempts. For the same question, a model may invoke different tools, in different orders, with different parameterizations. Sometimes it selects the correct tool chain; more often, it does not. Such variability is not meaningful evidence of robustness, especially in financial contexts where a single incorrect tool call may lead to irreversible monetary loss.
>
> A concrete example illustrates this clearly. Consider the query: “What’s the USDC/WETH pool price of the Uniswap v3 0.05 USDC/WETH pool on mainnet at block height 22636495?” When evaluated with five samples, only one of DeepSeek V3.1’s traces produced the correct answer. In this successful attempt, the model invoked the block query tool, then the data query tool, and finally the code execution tool to compute the price. This is the correct tool chain. The other four attempts failed to progress beyond the early steps, often choosing inappropriate tools or misinterpreting intermediate results, highlighting the brittleness of the model’s default reasoning process.
>
> We will include full multi-run tool-call histories in the revised appendix. These traces make it evident that aggregation obscures independent hallucinations rather than correcting them. The observed improvements under pass@5 therefore reflect stochastic variance rather than reliable reasoning.
> ***
> ## Comment 7
> We thank the reviewer for pointing out this ambiguity on tool usage. The two statements refer to different analytical levels. At a population level (Figure 4), stronger models tend to call slightly more tools on average, and these models also achieve higher accuracy; this produces a correlation between “more tool calls” and “higher model performance.” However, when we examine per-task behavior across more than 30,000 tool-enabled samples, a different picture emerges. For an individual question, accuracy is highest when the model answers with two to three tool calls, additional calls show diminishing returns. Thus, tool count primarily reflects task difficulty rather than improved performance. Our updated analysis shows that additional tool calls do not resolve the dominant failure modes, so repeated calls often reproduce the same mistakes. We will revise the paper to make this distinction explicit and ensure the narrative matches the empirical findings. More related analysis data is available at https://anonymous.4open.science/r/ICLR-AD05/caia/analysis/tool_calls/tool_calls_analysis.md
> ***
> ## Comment 8
> Finally, regarding the concern about ethical risks, especially the possibility that the benchmark may accelerate the development of more capable adversarial agents, we appreciate this important insight. We will clearly state in the revised manuscript and license terms that CAIA is intended to evaluate agent robustness, not to train or encourage manipulative or adversarial behaviors. We will also substantially tone down any implications that results extend to domains outside crypto without further evidence. This ethical framing will be reinforced through explicit disclaimers, greatly improving the responsible positioning of the benchmark.
> ***
> Thank you again for your thoughtful and generous review. Your feedback meaningfully strengthens both the rigor and responsibility of our work.

---

### Author Response · Authors · 2025-11-22
**General Comment**

We sincerely thank all reviewers for their thoughtful and engaged feedback. Across the reviews, we were encouraged by the consistent recognition of the paper’s core strengths. **Reviewers highlighted the importance of evaluating agent behavior in a real, adversarial, and economically meaningful domain.** This environment naturally reveals vulnerabilities in tool use, reasoning reliability, and misinformation resilience in ways synthetic benchmarks often cannot. Several reviewers also emphasized the value of grounding the benchmark in real user questions and pairing them with verifiable on-chain ground truth, which offers a uniquely transparent and reproducible evaluation setting. We are grateful as well for the positive reception of the human baseline, which multiple reviewers noted provides a practically meaningful reference point for interpreting model performance. More broadly, we appreciate **the acknowledgement that the benchmark is well-motivated and aligned with a wider effort to assess agents in high-stakes, real-data environments**.
***
### Common Concerns
At the same time, the reviews surfaced a few common areas where additional clarity would strengthen the paper. Most of the feedback centered around three themes **which have been effectively addressed in our revision**:
- methodological specificity, which includes prompt templates, agent orchestration, and tool-use mechanics
- transparency in dataset construction and curation
- more precise, measured framing of the benchmark’s scope and claims.
***
### Key Updates
In the revised manuscript, we have effectively addressed these points by:
 - Adding comprehensive methodological details, including unified prompt formats, tool-call scaffolding, and representative agent workflows.
 - Providing task examples across categories, illustrating both correct solutions and common failure modes.
 - Clarifying the dataset pipeline, including contributor recruitment, expert validation, LLM-assisted preprocessing, and strict answerability checks.

***
We thank the reviewers for highlighting where further clarity and caution were appropriate. Their feedback has substantially improved the rigor and transparency of the paper. We are grateful for the shared view that evaluating agent robustness in real, adversarial, economically meaningful settings is an important step for this area, and we hope the revised version presents CAIA as a careful, well-bounded contribution to that effort.

---

### Author Response · Authors · 2025-12-04
**General Comment to AC**

We are genuinely grateful for the reviewers regarding their helpful feedback. In the revised submission, we corrected the errors identified in the figures and addressed ambiguous or unclear statements in the main text. We also substantially expanded the appendix in direct response to reviewer suggestions. The updated appendix now includes:

- Additional experiments and uncertainty analysis

- A more detailed description of the experimental framework

- Example prompt template and concrete task walkthroughs

- The complete tool catalogue

- Formal definitions of all CAIA task categories with representative examples

- An expanded discussion of the adversarial threat model and real-world information environment

- A comprehensive description of the dataset construction and curation pipeline

- An explicit ethics section covering participant recruitment and study procedures

We wish these revisions improve transparency, resolve reviewers' concerns, and strengthen the methodological clarity of the paper.

---

### Note · Program_Chairs · 2026-01-17
**Submission Desk Rejected by Program Chairs**

The following references in this submission do not refer to real documents and/or have major errors in bibliographic information:

 Rui Zhang, Rui Xue, and Ling Liu. Challenges and opportunities in blockchain data management. IEEE Transactions on Knowledge and Data Engineering, 33(4):1451-1468, 2021.